# The TGF-β mimic TGM4 achieves cell specificity through combinatorial surface co-receptor binding

Shashi P Singh [1,6], Danielle J Smyth[1,7], Kyle T Cunningham [1], Ananya Mukundan [2], Chang-Hyeock Byeon[2], Cynthia S Hinck[2], Madeleine P J White[1,8], Claire Ciancia[1], Natalia Wąsowska[1,9], Anna Sanders [1], Regina Jin [1,10], Ruby F White[1], Sergio Lilla [3], Sara Zanivan[3,4], Christina Schoenherr [3], Gareth J Inman [3,4], Maarten van Dinther[5], Peter ten Dijke [5,11], Andrew P Hinck [2,11] & Rick M Maizels [1,6,11 ✉]

## Abstract

The immunoregulatory cytokine TGF-β is pleiotropic due to the near-ubiquitous expression of the TGF-β receptors TβRI and TβRII on diverse cell types. The helminth parasite *Heligmosomoides polygyrus* has convergently evolved a family of TGF-β mimics (TGMs) that bind both these receptors through domains 1–3 of a 5-domain protein. One member of this family, TGM4, differs from TGF-β in acting in a cell-specific manner, failing to stimulate fibroblasts, but activating SMAD phosphorylation in macrophages. Primarily through domains 4 and 5, TGM4 interacts with multiple co-receptors, including CD44, CD49d (integrin α4) and CD206, and can up- and downmodulate macrophage responses to IL-4 and lipopolysaccharide (LPS), respectively. The dependence of TGM4 on combinatorial interactions with co-receptors is due to a moderated affinity for TβRII that is more than 100-fold lower than for TGF-β. Thus the parasite has elaborated TGF-β receptor interactions to establish cell specificity through combinatorial *cis*-signalling, an innovation absent from the mammalian cytokine.

**Keywords** Agonist; Antagonist; Cytokine; Helminth; Receptor
**Subject Categories** Immunology; Microbiology, Virology & Host Pathogen Interaction; Signal Transduction

## Introduction

Many infectious agents exploit the pivotal host immunoregulatory pathway driven by transforming growth factor-β (TGF-β) (Maizels, 2021; Reed, 1999). In the case of helminth worm parasites, their fecundity and longevity depend upon a dampened host immune system, in some cases muted by immune-suppressive regulatory cells induced by cytokines such as TGF-β. Hence, it was remarkable to discover a helminth, *Heligmosomoides polygyrus*, that has convergently evolved a functionally active, but structurally unrelated, mimic of TGF-β (named TGM1) that binds strongly to mammalian plasma membrane TGF-β receptors (Johnston et al, 2017; Maizels and Newfeld, 2023; Mukundan et al, 2022). TGM1 acts as a fully functional activator of the TGFβ signalling pathway, downregulating inflammation in mouse models (Chauché et al, 2022; Redgrave et al, 2023; Smyth et al, 2021; Smyth et al, 2023) and inducing the differentiation of both mouse and human immuno-suppressive regulatory T cells (Tregs) through the canonical transcription factor Foxp3 (Cook et al, 2021; White et al, 2021). The ability of this parasite to drive Treg differentiation may therefore be explained by its production of TGM1 as central part of a strategy to evade host immunity (White et al, 2020).

TGF-β activates cells through a heteromeric receptor complex composed of two transmembrane serine/threonine kinases, TβRI (activin receptor-like kinase 5, ALK5) and TβRII (Hinck et al, 2016); its mode of binding is to first ligate TβRII, forming a complex that recruits and phosphorylates TβRI (Groppe et al, 2008). In contrast, TGM1 was found to independently bind both receptor subunits, with a particularly high affinity for TβRI (Johnston et al, 2017). TGM1 is comprised of five modular domains distantly related to the complement control protein (CCP) or Sushi protein family, with Domains domains 1 and 2 (D1-2) binding TβRI, while domain 3 (D3) binds TβRII (Mukundan et al, 2022). Thus, loss of any of domains 1–3 completely ablates activity of TGM1, confirming that binding to both receptor subunits is required for signal transduction (Smyth et al, 2018).

Recently, we ascertained that domains 4 and 5 (D4-5) of TGM1 confer an additional binding specificity, for the cell surface adhesion receptor CD44, which potentiates activation of cells through the TGF-β pathway (van Dinther et al, 2023). CD44 is

[1]Centre for Parasitology, School of Infection and Immunity, University of Glasgow, Glasgow G12 8TA, UK. [2]Department of Structural Biology, University of Pittsburgh, Pittsburgh, PA, USA. [3]Cancer Research UK Scotland Institute, Glasgow G61 1BD, UK. [4]School of Cancer Sciences, University of Glasgow, Glasgow, UK. [5]Oncode Institute and Department of Cell and Chemical Biology, University of Leiden, Leiden, The Netherlands. [6]Present address: Department of Biological Sciences, Birla Institute of Technology and Science, Pilani, Rajasthan 333031, India. [7]Present address: Division of Cell Signalling and Immunology, University of Dundee, Dundee, UK. [8]Present address: Gillies McIndoe Research Institute, Wellington, New Zealand. [9]Present address: Leiden University Medical Center, Leiden, The Netherlands. [10]Present address: Center for Cancer Research, Medical University of Vienna, Vienna, Austria. [11]These authors contributed equally: Peter ten Dijke, Andrew P Hinck, Rick M Maizels. ✉E-mail: rick.maizels@glasgow.ac.uk

widely expressed on hematopoietic cells, as well as some stem cell populations, interacting with extracellular matrix components such as hyaluronic acid (Ponta et al, 2003). It is prominent in immune cell types, upregulated in memory/effector T cells (Baaten et al, 2012), and includes a cytoplasmic domain capable of signal transduction. Hence, it was suggested that TGM1 has evolved to preferentially target CD44[+] immune cells for modulation during *H. polygyrus* infection (van Dinther et al, 2023).

*H. polygyrus* expresses a set of proteins related to TGM1, forming a multi-gene family of at least ten members with up to seven CCP-like domains (Smyth et al, 2018). Among these homologues, TGM2 and TGM3 with 93–100% identity in D1–3 showed similar functional activity to TGM1 activating the TGF-β pathway in a fibroblastic reporter cell line (Smyth et al, 2018). However, in the same assay, TGM4 was found to be inactive, despite the high sensitivity towards TGF-β/SMAD intracellular activity of the reporter cells for TGF-β/SMAD activation (Tesseur et al, 2006) and 80.2% amino acid identity to TGM1 (Fig. 1A,B).

We therefore performed a more detailed investigation of TGM4, which we now report differs from TGM1 in affinity for each TβR, and interacts with a wider range of cell surface co-receptors, resulting in a higher level of cell specificity that targets myeloid cells rather than fibroblasts and T lymphocytes. Taken together, these results highlight the importance of co-receptors, in addition to the TβRs, in delivering cell-specific signals via *cis*-interaction of cell surface receptors. This remarkable TGM gene family encoding secreted proteins comprising multiple discrete domains has thereby gained the ability, unlike TGF-β itself, to selectively target different host cell populations.

# Results

## Selective activation of T cells and macrophages by TGM4

The novel TGF-β mimic (TGM) family from *H. polygyrus* was first identified by activation of the TGF-β-responsive fibroblast-derived reporter cell line MFB-F11, and subsequently shown to induce the transcription factor Foxp3 in naive murine splenic T cells (Johnston et al, 2017; Smyth et al, 2018). As reported previously (Johnston et al, 2017; Smyth et al, 2018), TGM4 showed no activity in the same assay (Fig. 1A). However, a small but significant signal was found with a fibroblastic (NIH 3T3) CAGA$_{12}$ SMAD3-responsive luciferase transcriptional reporter cell line (Fig. 1B). We therefore examined whether TGM4 could drive Foxp3 in mouse T cells and found positive induction albeit at lower efficacy than TGM1 (Fig. 1C), while another family member that was inactive on MFB-F11 cells (TGM7, (Smyth et al, 2018)) was also negative for Foxp3 induction. Hence, TGM4 presented a uniquely discordant activity that we investigated further.

To confirm that TGM4 was, like TGM1, acting through the SMAD signalling pathway in T cells, cultures were supplemented with the small molecule inhibitor SB431542 which blocks the kinase activity of ALK4, 5 (TβRI) and 7 (Inman et al, 2002). As shown in Fig. 1C, while TGM4 was less potent than TGM1 in inducing Foxp3, both ligands were fully suppressed by the addition of SB431542, as previously established for TGM1 (Johnston et al, 2017).

We then probed different cell types for TGM1- and TGM4-stimulated phosphorylation of SMAD2 that is immediately downstream of the TβRI kinase in the signalling pathway. Fibroblast,

macrophage and T-cell lines were incubated with these ligands, and mammalian TGF-β, for 60 min, then lysed and analysed by western blot with anti-SMAD2/3 and anti-phospho-SMAD2 (p-SMAD2) antibodies. As shown in Fig. 1D,E TGM4-stimulated MFB-F11 fibroblasts showed only low levels of response. However, in the murine EL4 T-cell line, and in macrophage lines (J774A.1 and RAW264.7) all three ligands were equally active. We also tested a human hepatoma cell line, HepG2, which responded only to TGF-β (Fig. 1D).

In addition to cell lines, we investigated responses of mouse primary hematopoietic cells, measuring SMAD2 phosphorylation 1 h following stimulation with the different ligands. When CD4[+] murine splenic T cells were stimulated with TGM4, responses were much weaker than with TGF-β or TGM1, and indeed did not attain statistical significance (Fig. 1F). As similar cells were capable of Foxp3 induction after 72 h co-incubation with TGM4 (Fig. 1C), the possibility was raised that activation follows a slower time course, as indeed observed for TGM1 compared to TGF-β (White et al, 2021). To test this, we employed imaging flow cytometry to measure nuclear localisation of SMAD2/3 in splenic T cells at 1 and 16 h post-stimulation; although at the earlier time point TGM4-stimulated T cells were at baseline values (Fig. EV2A), by 16 h they were elevated and comparable to cells activated with the other ligands (Fig. EV2B). Thus, while TGM4 has a relatively subdued ability to activate TGF-β signalling in primary T cells, it is sufficient to induce measurable responses over a 16–72 h time frame.

We similarly analysed SMAD activation in mouse bone marrow-derived macrophages; in this subset, responses to each ligand were comparable, with TGM4 also inducing a significant level of pSMAD2/3 measured by western blot analysis (Fig. 1G) and SMAD2/3 nuclear localisation (Fig. EV2C) within 1 h of stimulation. In additional analyses, TGM4 was found to lack activity on other mouse epithelial (NM18) and fibroblast (NIH 3T3) cell lines (Fig. EV2D–F), but did induce SMAD2 phosphorylation in dendritic cells (DCs), represented both as mouse cell lines (MuTu and D1, Fig. EV2G,H) and bone marrow-derived DCs (Fig. EV2H). Hence, TGM4 displays a strong predilection for cells of the myeloid lineage.

## TGF-β receptor binding by TGM4

We next tested whether activation of the SMAD signalling pathway by TGM4 could be induced by the same domains (D1-3) that are required for activation by TGM1 (Smyth et al, 2018). Fibroblast and macrophage cell lines incubated with full-length (D1-5) or truncated (D1-3 or D4–5) portions of TGM1 and TGM4 were probed for SMAD2 phosphorylation. As shown in Fig. 2A, only full-length or D1-3 of TGM1 elicited a p-SMAD2 response in MFB-F11 fibroblasts, but neither full-length nor D1-3 of TGM4 did so. In contrast, when a macrophage cell line, RAW264.7, was tested both proteins drove p-SMAD2 (Fig. 2B). Notably, for both TGM1 and TGM4 activity was attenuated in forms lacking D4–5.

To investigate whether TGM4 differs from TGM1 in its interactions with TβRI and TβRII, we next used a system of endogenous expression of enhanced green fluorescent protein (eGFP)-TGM fusion proteins in transfected MFB-F11 fibroblasts, followed by GFP-TRAP pulldown of the ligand and any associated receptors. In this manner, we identified that TGM1 and TGM4, but not TGM7, form complexes with both TβRI (Fig. 2C,D) and TβRII (Fig. 2E,F). While TGM4 co-precipitated TβRI more strongly than

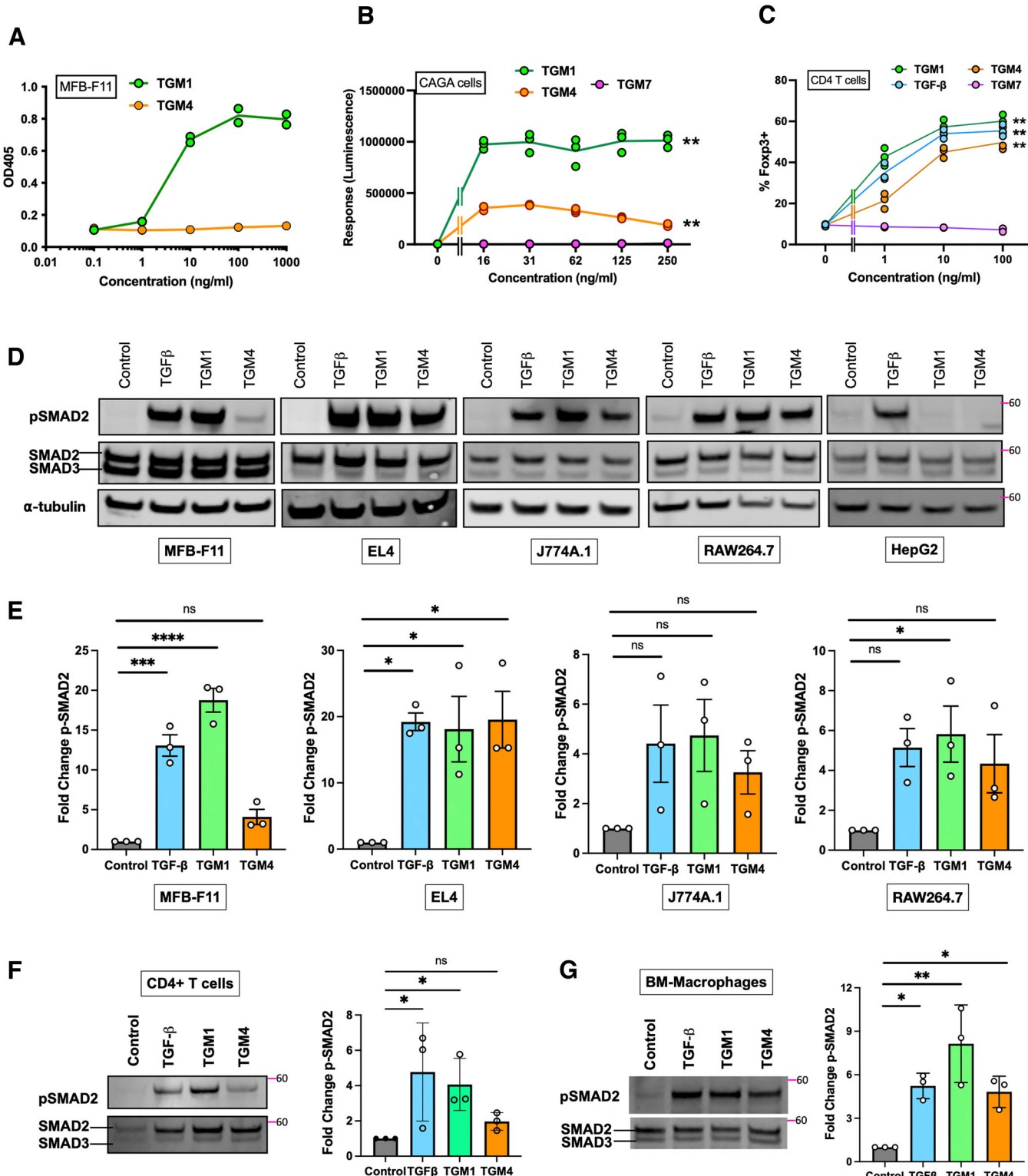

did TGM1, the interaction with TβRII was substantially weaker with TGM4 than with TGM1.

To ascertain whether TGM4 binds directly to each receptor chain, and to understand the respective receptor affinities of TGM1 and TGM4, surface plasmon resonance (SPR) measurements were performed to determine their binding to TβRI and TβRII. SPR analysis showed substantially higher affinity of TGM4 than TGM1 (Fig. 2G,H) for TβRI, 3–5 nM vs 90 nM, respectively (Table 1; Appendix Table S1). Notably, TGM4 binding to TβRI shows a faster on-rate and slower off-rate than does TGM1.

**Figure 1.   Differential activation of fibroblasts, T cells and macrophages by TGM4.**

(A) Response of MFB-F11 reporter fibroblasts to recombinant TGM1 and TGM4 proteins, measured by an enzymatic assay (OD405 nm) for release of alkaline phosphatase. Data shown are from 1 of 4 biological replicate experiments with similar results ($n = 2$). (B) Response of CAGA12 luciferase reporters to TGM1 and TGM4; a more distantly related family member, TGM7, is included as a negative control. Data shown are from 1 of 2 biological replicate experiments with similar results; mean ± SD ($n = 3$). (C) Induction of Foxp3 expression in mouse splenic CD4$^+$ T cells incubated with TGM1, TGM4, TGM7 or TGF-β. Data shown are from 1 of 2 biological replicate experiments with similar results ($n = 3$); mean ± SD. (D) SMAD2 phosphorylation in cell lines of MFB-F11 fibroblasts, EL4 T lymphocytes, J774A.1 RAW264.7 macrophages, and HepG2 cells stimulated with TGF-β, TGM1 and TGM4, measured by western blotting; upper row probed with anti-pSMAD2; lower row with anti-SMAD2/3 antibody. Images are from 1 of 3 biological replicate experiments. (E) Densitometric analyses of SMAD2 phosphorylation, as in (E), from all three biological replicate experiments, showing means ± SD. (F) SMAD2 phosphorylation in primary splenic CD4$^+$ T cells, assessed as in (D, E), by western blot in 1 of 3 biological replicate experiments (left panel) and by densitometric analyses of all three experiments (right panel), showing means ± SD. (G) SMAD2 phosphorylation in bone marrow-derived macrophages, assessed as in (D, E), by western blot in 1 of 3 biological replicate experiments (left panel) and by densitometric analyses of all three experiments (right panel), showing means ± SD. Data Information: Data in (B) analysed by two-way ANOVA with Tukey's correction; at 250 µg/ml, TGM1 vs TGM4 $P = 0.0021$; TGM1 vs TGM7 $P = 0.0014$; TGM4 vs TGM7 $P = 0.0029$. Data in (C) analysed by two-way ANOVA with Tukey's correction; at 100 µg/ml TGM1 vs TGM4 $P = 0.0937$; TGM1 vs TGM7 $P = 0.0011$; TGM4 vs TGM7 $P = 0.0064$; TGF-β vs TGM7 $P = 0.0011$. (E–G) Analysed by one-way ANOVA with Dunnett's multiple comparison tests. In (E), for MFB-F11 Control vs TGF-β $P = 0.0002$, Control vs TGM1 $P < 0.0001$; for EL4, Control vs TGF-β $P = 0.011$, Control vs TGM1 $P = 0.0169$ and for Control vs TGM4 $P = 0.0121$; for RAW264.7, Control vs TGM1 $P = 0.0392$. (F) Control vs TGF-β $P = 0.0364$, Control vs TGM1 $P = 0.0499$. In (G), Control vs TGF-β $P = 0.0222$, Control vs TGM1 $P = 0.0011$ and for Control vs TGM4 $P = 0.0360$. In all panels, ns = not significant, *$P < 0.05$, **$P < 0.01$, ***$P < 0.001$, ****$P < 0.0001$.

In contrast to the higher affinity of TGM4 for TβRI, binding to TβRII was found to be weak by SPR, at 116 µM, representing >100-fold lower affinity than TGM1 (Fig. 2I; Table 1; Appendix Table S1). Nuclear magnetic resonance (NMR) analysis showed small but significant shifts of several signals when $^{15}$N-labelled Domain 3 of TGM4 was titrated with increasing concentrations of TβRII (Fig. 2J), also indicative of relatively low binding affinity. As this raised the possibility that the physiological target of TGM4 is another type II receptor, we performed isothermal titration calorimetry (ITC) binding assays with both activin type II receptors (ActRII and ActRIIB) and with bone morphogenetic protein type II receptor (BMPRII), three major related receptors of this type. However, we found no evidence of direct binding to any of these receptors (Fig. EV3A–C). In addition, no interactions were observed between D2 and the activin type Ib receptor, ALK4 (Fig. 3EVD). We therefore concluded that the cognate receptors for TGM4 are TβRI (ALK5) and TβRII, similar to TGM1, although the two parasite proteins differ markedly, and reciprocally, in affinity for these receptors.

## CD44 binding by TGM4

TGM1 and -4 share the same 5-domain structure (Fig. EV1A); in the case of TGM1, we showed by truncation analysis that only D1–3 were essential for biological activity (Smyth et al, 2018); however, D45 enhance the potency of TGM1, binding to a cell surface co-receptor identified as CD44 (van Dinther et al, 2023). To determine if TGM4 also binds CD44, we expressed eGFP-tagged TGM1 and TGM4 in MFB-F11, RAW264.7 and HepG2 cells, followed by GFP-TRAP pulldown and western blotting analyses. As shown in Fig. 3A, a strong CD44 band was observed with both TGM1 and TGM4 in the fibroblast and macrophage cell lines, but not in hepatocytes. Quantification from replicate western blot experiments showed that TGM4 co-precipitated considerably more CD44 from RAW264.7 cells than did TGM1, and a similar trend was observed with MFB-F11 cells (Fig. 3B). In addition, higher TβRI levels were co-precipitated with TGM4 than TGM1 in RAW264.7 cells (Fig. 3A,C). In contrast, TβRII co-precipitation was significantly greater with TGM1 (Fig. 3A,D), consistent with the weak TGM4 binding to this receptor noted above. Taken together, these data suggest that the strength of interaction with

CD44 and both TβRs could be related to the differential activity of TGM4 on these two cell types.

To dissect interactions of TGM4 with each receptor at the domain level, eGFP fusions of truncated proteins were designed, expressing D1-3 or D4–5 of TGM1 and TGM4 in RAW264.7 cells. Lysates of RAW264.7 cells expressing each construct were immunoprecipitated using GFP-TRAP beads and analysed by western blotting. D1-3 constructs precipitated the TβRs, but not CD44, while when D4–5 was expressed, the converse was true (Fig. 3E), as recently reported for TGM1 (van Dinther et al, 2023).

To more precisely evaluate TGM4-CD44 interactions, the binding of TGM1 and TGM4 with recombinant human and mouse CD44 was measured using ITC. For these measurements, the D4–5 fragments of each TGM were employed, as this segment of TGM1 carries the CD44-binding capacity (van Dinther et al, 2023). As shown in Fig. 3F,G, both helminth proteins bound each CD44 molecule with similar affinities, which were determined to be in the 50–150 nM range (Table 1; Appendix Table S2).

To establish whether CD44 was essential for TGM4 activation of myeloid cells, we chose to study RAW264.7 macrophages as these can be modified by CRISPR-Cas9 gene editing; in this way we depleted CD44 expression (Fig. EV4A) and evaluated responses of these macrophages to increasing doses of TGM4. As shown in Fig. 3H,I, SMAD phosphorylation was reduced to background levels in macrophages lacking CD44, across a range of concentrations of TGM4 that elicited strong responses in CD44-sufficient cells.

## Preferential binding of TGM4 to myeloid cells

We recently reported that Alexa Fluor-488 (AF488)-labelled TGM1 binds strongly to the surface of MFB-F11 and EL4 cells, as measured by flow cytometry (van Dinther et al, 2023). We noted that TGM4 bound these, and other, cell lines with greater intensity than seen with TGM1 (Fig. EV4B). Preferential binding of TGM4 was most evident on two macrophage cell lines (J774A-1 and RAW264.7), while no binding was observed to the hepatocyte line, HepG2. Notably, the intensity of binding to J774 macrophages was significantly higher than each of the other immune-derived lines.

To test the relative strength of TGM1 and TGM4 interactions across a range of physiological cell types, we evaluated binding to primary peritoneal exudate cells from mice, by flow cytometry

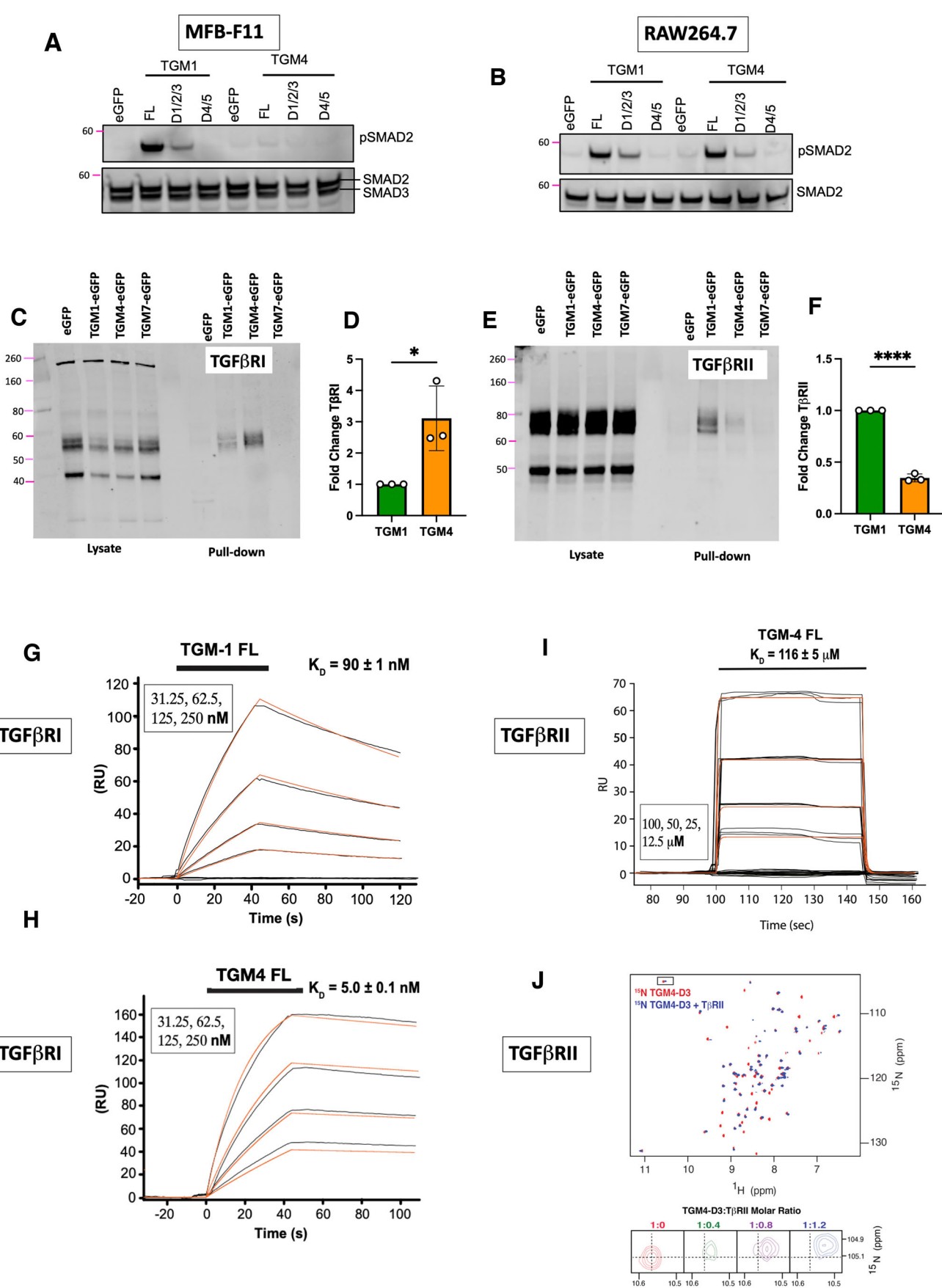

**Figure 2.   TGM4 domain and receptor binding analysis.**

(A, B) SMAD2 phosphorylation induced by D1-3 by both TGM1 and TGM4 when added to MFB-F11 fibroblasts (A) or RAW264.7 macrophage cell line (B). Serum-starved cells were incubated with conditioned medium from eGFP alone, TGM4 D1-5, D1-3 or D4–5 transfected cells for 1 h and analysed for pSMAD2 and SMAD2/3 by western blotting. Images are from 1 of 3 biological replicate experiments. (C) TβRI GFP-TRAP pulldown from MFB-F11 cells transfected with eGFP alone, or eGFP fused to TGM1, TGM4 or TGM7. (C) Whole-cell lysates shown on left-hand side of image, and anti-GFP immunoprecipitates (pulldowns) shown on right hand side, following western blot with anti-TβRI antibody. Image is from 1 of 3 biological replicate experiments. (D) Densitometric analyses of TβRI pulldowns, as in (C), from all three biological replicate experiments. Data shown as mean ± SD. (E) TβRII GFP-TRAP pulldown from MFB-F11 cells transfected with eGFP alone, or eGFP fused to TGM1, TGM4 or TGM7, as in (C). The image is from 1 of 3 biological replication experiments. (F) Densitometric analyses of TβRII pulldowns, as in (E), from all three biological replicate experiments. Data shown as mean ± SD and analysed by unpaired *t* test, ****$P < 0.0001$. (G, H) Surface plasmon resonance (SPR) sensorgrams of full-length TGM1 and TGM4 binding to biotinylated Avi-tagged TβRI immobilised on a streptavidin chip. Injections were performed as a twofold dilution series and are shown in black, with the orange traces over the raw data showing curves fitted to a 1:1 model. The black bars over the sensorgrams specify the injection period and the injection concentrations are indicated in the upper left. (I) SPR sensorgram of full-length TGM4 binding to biotinylated Avi-tagged TβRII immobilised on a streptavidin chip, with injection performed as above; injection period depicted by the black bar at top, and injection concentrations at bottom left. (J) NMR analysis of TGM4 D3 binding to TβRII; $^{15}$N-labelled D3 alone was at a concentration of 100 μM (red) and is overlaid with the $^1$H-$^{15}$N spectra of the same protein bound to 1.2 molar equivalents of unlabelled TβRII (blue). Expansion of intermediate titration points (1:0, 1:0.4, 1:0.8, 1:1.2 $^{15}$N-TGM4 D3:TβRII) of the boxed signals are shown in the lower panel. Data showing no interactions with other Type II receptors are presented in Fig. EV3. Data Information: Data in (D, E) analysed by unpaired Student's *t* test; in (D), $P = 0.0241$ and in (E), $P < 0.0001$. In both panels, *$P < 0.05$, ****$P < 0.0001$.

**Table 1.   Summary of ligand–receptor binding affinities.**

| TGM ligand | Measurement | TβRI | TβRII | mCD44 | hCD44 |
|---|---|---|---|---|---|
| TGM1-FL | SPR | 90 ± 1 nM | 0.61 ± 0.01 μM[a] | | |
| TGM4-FL | SPR | 5.0 ± 0.1 nM | 116 ± 2 μM | | |
| TGM1-D45 | ITC | | | 30 (9, 68) nM[b] | 122 (73, 195) nM[b] |
| TGM4-D45 | ITC | | | 20 (7, 41) nM[b] | 78 (40, 136) nM[b] |

Affinity measurements of full-length (FL) TGM1 and TGM4 for TβRI and TβRII were measured by Surface Plasmon Resonance (SPR); affinities of D4-5 of the two proteins for CD44 of mouse (mCD44) and human (hCD44) were measured by Isothermal Titration Calorimetry (ITC).
[a]Value previously reported by Mukundan et al (Mukundan et al, 2022).
[b]Values in parenthesis represent estimated upper and lower bounds by the Sedphat fitting algorithm.

using AF594-labelled proteins. Both ligands showed extensive binding to CD3$^+$ T cells and CD11b$^+$F4/80$^+$MHC-II$^{low/−}$ tissue-resident macrophages (Fig. 4A), and in each case staining was more intense by TGM4 compared to TGM1 (Fig. 4B). Staining of T lymphocytes was quite heterogenous, while 100% of macrophages were bound by TGM4.

Although TGM4 showed more intense staining, it is possible that differences in efficiency of labelling or protein stability affected by coupling Alexa Fluor to exposed lysine residues could be responsible. However, when we co-stained with both proteins, coupling TGM1 to AF594 and TGM4 to AF488 and testing on peritoneal resident macrophages (Fig. 4C), TGM4 outcompeted TGM1, reducing the TGM1 signal by ~40%, while the TGM4 signal was only modestly reduced (~15%) in the presence of TGM1 (Fig. 4D). Similar inhibition was observed on staining other peritoneal cell subsets (Fig. EV4C). Notably, TGM1 and TGM4 doubly-stained cells gave a diagonal indicating that the two ligands bind similar populations of host cells. Taken together, these data demonstrate that TGM4 is more avid than TGM1 in surface binding to immune system cells, with a strong affinity for myeloid cells.

To examine the role of CD44 in surface interactions, we tested CD44-sufficient and -deficient RAW264.7 macrophages by flow cytometry with AF488-labelled full-length and truncated TGM4 proteins comprising D1-3 or D4–5. As shown in Fig. 4E, we found that in the absence of either CD44, or D4–5, cell surface binding is effectively abolished. Thus, on CD44-sufficient RAW264.7 cells (upper row of Fig. 4E), binding of full-length TGM4, or of D4–5,

correlates closely with expression of CD44, while D1-3 shows only a low level of fluorescence; in the absence of CD44 expression (lower row) a similar residual level of binding is observed in full-length TGM4 and D1-3, suggesting this represents interactions with the TβRI/II proteins. It was also noted that the intensity of binding (MFI) of D4–5 was attenuated compared to full-length TGM4 (Fig. 4F,G). Similar data were obtained with MFB-F11 cells with intact or deleted CD44 expression (Fig. EV4D,E). As with RAW264.7 cells, D4-5 binding to MFB-F11 fibroblasts was lower than full-length TGM4, suggesting that D1-3 may contribute to the overall binding to CD44. A similar involvement of D1-3 in optimal CD44 binding was also indicated by the ability of the unlabelled D4–5 to diminish but not fully inhibit the binding of full-length AF488-labelled TGM4 to RAW264.7 cells (Fig. 4H,I).

In the case of TGM1, D1 was found to contribute to the ability of D2 to bind TβRI, (Mukundan et al, 2022), explaining earlier data that D1–3 were all essential for biological activity in the MFB-F11 assay (Smyth et al, 2018). A similar truncation analysis for TGM4 was performed using RAW264.7 macrophages, evaluating pSMAD2/3 activation, indicating a parallel dependency, as in the absence of any of the first 3 domains, signalling was ablated, while in the absence of D4 and/or D5, an attenuated level of signalling could be detected (Fig. EV4F).

## TGM4 binds additional co-receptors

To better understand why TGM4 binds cells more avidly than TGM1 despite a similar affinity for CD44 within D4–5, we investigated the

                                                                  

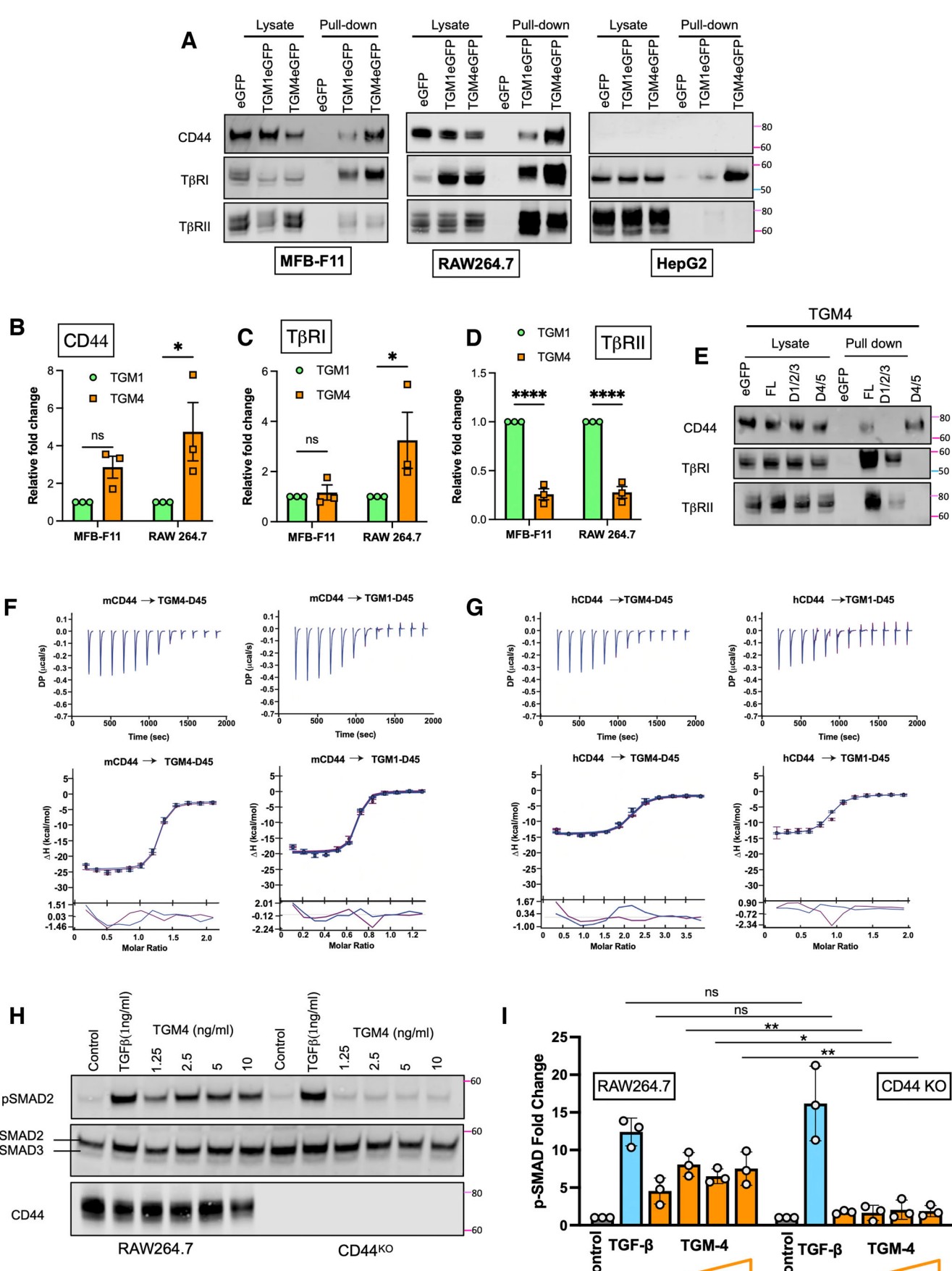

◄ **Figure 3. TGM4, like TGM1, binds CD44.**

(A) eGFP trap pull-down and western blotting analyses of MFB-F11, RAW264.7 and HepG2 cells transfected with eGFP alone, or with eGFP-TGM1 or TGM4 fusions; whole-cell lysates shown in the left-hand side of each panel, and anti-GFP immunoprecipitates (pulldowns) shown on the right; western blots were probed with antibodies to CD44, TβRI, and TβRII. Images are from 1 of 3 biological replicate experiments. (B–D) Densitometric analysis, from all three biological replicate experiments, of pull-down proteins CD44 (B), TβRI (C), and TβRII (D) from MFB-F11 or RAW264.7 cells expressing TGM1 or TGM4. Data are shown as fold change relative to TGM1 (B, C) or TGM4 (D) that have been respectively normalised to 1. Data presented as mean ± SD of all biological replicates. (E) eGFP trap pulldown and western blotting RAW264.7 cells as in (A), with TGM4 full-length (FL) and truncated constructs D1-3 and D4-5. Images are from 1 of 3 biological replicate experiments. (F, G) ITC binding isotherms of TGM4 D4-5 binding to murine CD44 (F, G, respectively). Two technical replicate measurements, depicted in blue and purple, were globally fit to a single binding isotherm. (H, I) TGM4 activation of pSMAD signalling in RAW264.6 macrophages is dependent on CD44 expression, shown as western blot images from 1 of 3 biological replicate experiments (H), and densitometric data from all three biological replicate experiments, with increasing doses of TGM4 corresponding to doses of 1.25, 2.5, 5 and 10 μg/ml (I); in (I) TGF-β is shown to activate in a CD44-independent manner. Data Information: Data in (B–D, I) analysed by two-way ANOVA with Sidak's multiple comparison test. In (B), TGM1 vs TGM4 in RAW264.7 cells, $P = 0.025$; in (C), TGM1 vs TGM4 in RAW264.7 cells, $P = 0.049$; in (D), TGM1 vs TGM4 in both MFB-F11 and RAW264.7 cells, $P < 0.0001$; in (I), at 2.5 μg/ml $P = 0.002$; at 5 μg/ml $P = 0.046$; and at 10 μg/ml $P = 0.002$. In all panels, ns = not significant, $*P < 0.05$, $**P < 0.01$, $****P < 0.0001$.

possibility that TGM4 interacts with additional co-receptors not recognised by TGM1, by adding exogenous biotin-labelled TGM proteins to cell suspensions and performing pulldowns with streptavidin beads. Pulldown of TGM4-binding proteins from splenocytes revealed, in addition to CD44 and TGM4 itself, four more candidates: CD49d (integrin α4), CD72, CD206 (Mrc1) and Lirb3 (Fig. 5A). While CD44 was evident in pulldowns from MFB-F11 cells, none of these other candidates were apparent (Fig. 5B). We also performed similar procedures on the macrophage cell line J774, confirming that TGM4 interacted with CD49d and CD72, in addition to CD44 (Fig. 5C). In addition, in all cells probed with TGM4, neuropilin-1 (NRP-1) was identified. Notably, parallel analyses of the same three cell populations with TGM1 failed to show interactions with these CD49d, CD72 or NRP-1 (Fig. 5D–F).

Taken together, these data demonstrated that TGM4 associates not only with the CD44 co-receptor, but also CD49d and CD72 which were not detected on fibroblasts, and with NRP-1 that was expressed in all three cell types studied (Fig. EV5A).

## CD44-dependent and independent co-receptors

In co-precipitation experiments, partner proteins may interact with different members of a complex. To ascertain whether TGMs directly bind co-receptor proteins, or do so in conjunction with CD44, we performed pull-down experiments in RAW264.7 macrophages with unaltered or deleted *Cd44* expression, incubated with exogenous biotinylated TGM proteins followed by streptavidin affinity purification. In CD44-sufficient cells, CD49d, CD206 and NRP-1 were all detectable in precipitates from cells incubated with biotinylated TGM4, but not TGM1 (Fig. 5G); the loss of CD44 from RAW264.7 cells ablated CD206 and NRP-1 detection, but CD49d remained present in the TGM4 pull-down. Hence although CD206 and NRP-1 are found only in TGM4 complexes, their presence is dependent on CD44. A parallel result was observed in MFB-F11 cells, in which CD49d is not expressed, but NRP-1 is present; as with RAW264.7 cells, NRP-1 is precipitated only in wild-type CD44-sufficient cells (Fig. EV5B).

We next asked whether the CD44-binding domains D4–5 are required for association with these three co-receptors; the same streptavidin pull-down system was used with wild-type RAW264.7 macrophages incubated with biotin-labelled full-length (D1-5), D1-3 or D4–5 TGM4. CD206 and NRP-1 were found only in full-length and D4–5 constructs, consistent with them being dependent on CD44. In contrast, CD49d was associated with D1-3 proteins, which also

bind the two TGF-β receptors, albeit the binding is weaker than with full-length TGM4 (Fig. 5H).

CD49d and NRP-1 knockout RAW264.7 macrophages were then tested to determine if either gene was required for signalling in response to TGM4. These cells, and unmodified controls, were stimulated with TGM4 for 30 min, and cell lysates probed for SMAD phosphorylation by western blot. However, in neither case did gene deletion reduce responses as measured by pSMAD2 levels (Fig. EV5C,D). Moreover, when KO cells were tested by flow cytometry, no diminution of binding by TGM4 was noted (Fig. EV5E).

## TGM4 efficiently modulates macrophage function

To test whether the preferential targeting of myeloid cells by TGM4 has functional consequences, we tested its impact on the in vitro response of bone marrow-derived (BMDMs) macrophages to key type 1 (lipopolysaccharide, LPS) and type 2 (IL-4) stimulation. As shown in Fig. 6, the inflammatory responses of BMDM cells, as measured by TNF (Fig. 6A), interleukin (IL)-6 (Fig. 6B) and IL-1β (Fig. 6C) secretion, were inhibited to an equal degree by TGM1, TGM4 and TGF-β, although inhibition of IL-1β release was relatively muted. We next examined the macrophage response to IL-4, which is characterised by induction of M2 genes such as Arginase-1 (Boutard et al, 1995), chitinase-like protein Chi3L3 (Ym1) and resistin-like molecule α (RELMα) (Loke et al, 2002; Martinez et al, 2009). When TGF-β or TGMs are co-administered with IL-4, Arginase-1 (Fig. 6D) and Chi3L3 (Fig. 6E) expression are each markedly amplified while RELMα secretion is inhibited (Fig. 6F), in an uncoupling of these markers that are often considered co-ordinately produced.

Finally, to question whether TGM4 would have similar effects on primary macrophages in vivo, we administered recombinant proteins into the peritoneal cavity of BALB/c mice and recovered myeloid populations 24 h later. We focused on the resident F4/80^hi, MHC-II^lo peritoneal macrophage population, using flow cytometry, co-staining for type 2 macrophage markers, revealing a significant increase in Arginase-1 expression (Fig. 6G), and repression of RELMα (Fig. 6H) that had been observed in bone marrow-derived macrophages (BMDMs) in vitro, with enhancement of Arginase-1 significantly attenuated in the absence of D4-5-CD44 interactions. Further examination of surface markers revealed a significant reduction in CD86 expression by TGM4 within the resident large peritoneal macrophage (LPM) subset (Fig. 6I), again contingent upon the presence of the co-receptor binding D4-5.

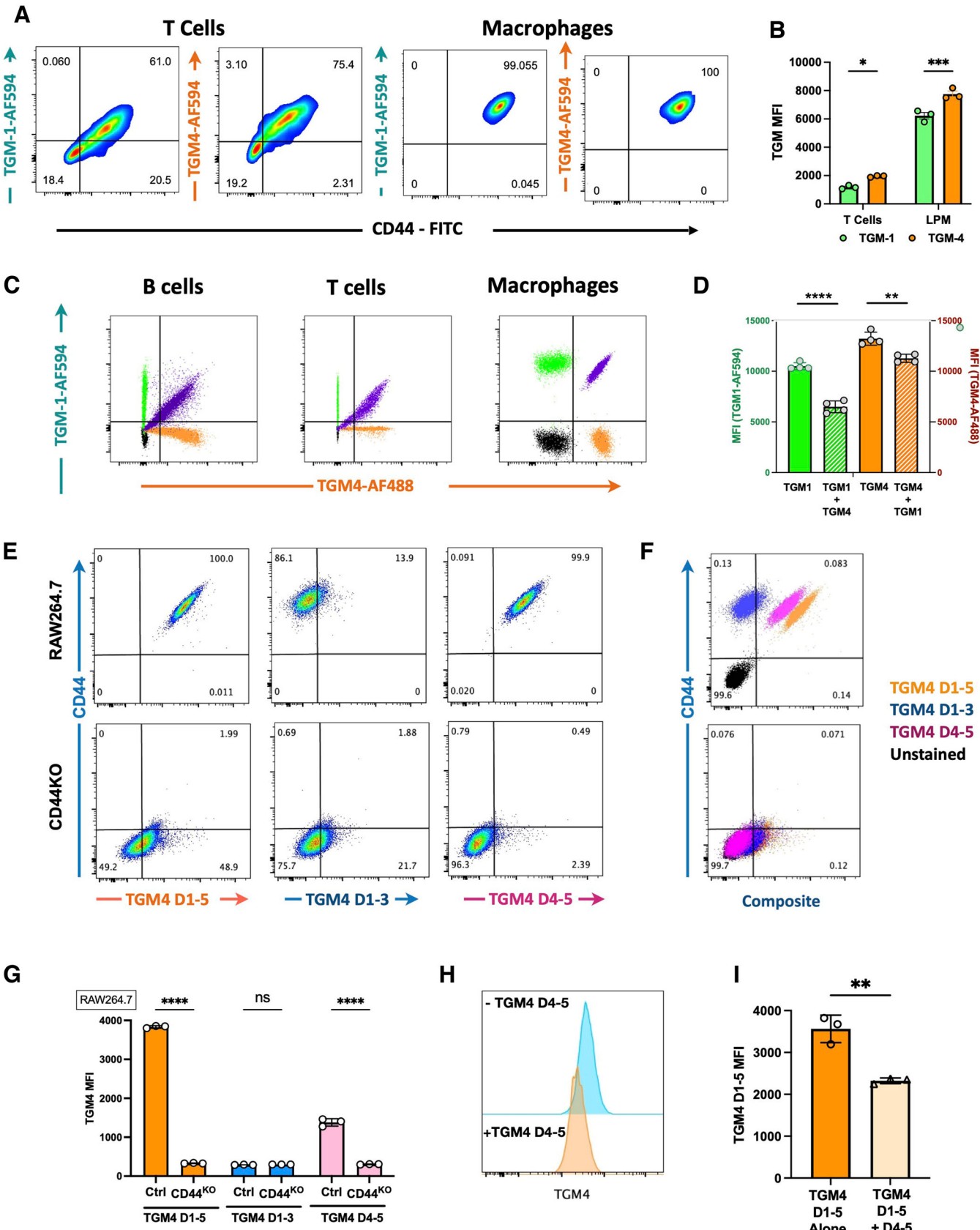

**Figure 4.  TGM4 binding to host immune cells.**

(A) Co-staining of CD44 and Alexa Fluor 594 (AF594)-labelled TGM1 or TGM4 to peritoneal CD3[+] T cells, and CD11b[+]F4/80[(high)] MHC-II[(low)] large peritoneal macrophages, measured by flow cytometry. Percentages of the target populations in each quadrant from 1 of 2 biological replicate experiments are shown. (B) Mean fluorescent intensity (MFI) for AF594-labelled TGM1 or TGM4 binding to peritoneal T cells and large peritoneal macrophages. Data shown are from 1 of 2 biological replicate experiments, with $n = 3$, showing mean ± SE. (C) Co-staining of AF594-labelled TGM1 and AF488-labelled TGM4 to peritoneal CD19[+] B cells, CD3[+] T cells and large peritoneal macrophages. Plots are superimposed from the indicated cell populations stained with TGM1 (cyan), TGM4 (orange), both TGM1 and TGM4 (purple) or unstained control (black). Data shown are from 1 of 2 biological replicate experiments. (D) Quantification of staining of large peritoneal macrophages by AF594-TGM1 or AF488-TGM4 in the absence of the presence of TGM4 or TGM1, respectively, as measured by MFI; the combination of TGM1 and TGM4 reduced the TGM1 signal by 38.2% and the TGM4 signal by 14.7%. Data shown are from 1 of 2 biological replicate experiments, with $n = 4$, showing mean ± SE. (E) Flow cytometric analysis of TGM4 binding to RAW264.7 wild-type and CD44-deficient cells, probed with full-length TGM4 D1-5, and truncated constructs D1-3 and D4–5. Data shown are from 1 of 3 biological replicate experiments. (F) Superimposition of the 3 datasets, with full-length TGM4 D1-5 (orange), D1-3 (dark blue), D4–5 (magenta), together with unstained control (black). Data shown are from 1 of 3 biological replicate experiments. (G) MFI in all three biological replicate experiments comparing binding of the TGM4 full-length and truncation constructs to wild-type and CD44-deficient RAW264.7 cells. Data are presented as mean ± SD. (H, I) Partial inhibition of AF AF488 FL TGM4 binding to RAW264.7 in the presence of unlabelled TGM4 D4–5 shown as exemplar histogram (G) and data from three technical replicate samples (H). Data are presented as mean ± SD. Data Information: Data in (B, D, G) analysed by two-way ANOVA with Sidak's multiple comparison test. Data in (I) analysed by unpaired Student's $t$ test. In (B), TGM1 vs TGM4 in T cells, $P = 0.0200$; TGM1 vs TGM4 in LPM cells, $P = 0.0003$; in (D), TGM1 + TGM4 vs TGM1 alone, $P < 0.0001$; TGM4 + TGM1 vs TGM4 alone, $P = 0.0014$; in (G) Control vs CD44 KO with TGM4 D1-5 and with TGM4 D4–5 $P < 0.0001$; in (I) TGM4 D1-5 alone vs TGM4 D-15 + D4–5, $P = 0.003$. In all panels, ns = not significant, *$P < 0.05$, **$P < 0.01$, ***$P < 0.001$, ****$P < 0.0001$.

## Discussion

Host immunity has been a powerful force for the evolution and innovation of pathogen escape mechanisms. The TGM protein family of *H. polygyrus* has acquired a novel elaboration to confer cell-selectivity for mimics of a pivotal immune system cytokine, TGF-β. In so doing, this parasite may have adopted a successful strategy for immune evasion that is shared, in a different fashion, by many diverse infectious agents (Maizels, 2021). More strikingly, the TGM proteins have evolved a pattern of *cis*-signalling that goes beyond ligation and recruitment of the two TGF-β receptor subunits, and encompasses co-receptors associated with specific target cell types. The ability to bind multiple cellular receptors is made possible by the modular structure of TGMs comprising 3 to 7 homologous non-identical domains that ligate different specificities.

The homology domains of TGM, are all distantly related to the CCP or Sushi family (Pfam00084), modified by short insertions that evidently allow evolutionary flexibility and the ability to bind novel interaction partners (Mukundan et al, 2022). Both TGM1 and TGM4 are 5-domain proteins with D1-2 binding TβRI and D3 binding TβRII. Although the corresponding domains of the two proteins are highly conserved (86–90% amino acid identity), we find sharply contrasting binding affinities, with TGM4 being tenfold stronger for TβRI, yet 100-fold weaker in binding TβRII.

These differences may account for the inability of TGM4 to activate fibroblasts, while retaining some ability to induce T-cell expression of Foxp3, the canonical transcription factor of suppressive Tregs which are expanded during *H. polygyrus* infection (Finney et al, 2007; Smith et al, 2016). However, TGM4 is markedly more active on myeloid cells than other cell types, which is significant given the pivotal role of macrophages and neutrophils in priming and mediating protective immunity to *H. polygyrus* (Anthony et al, 2006; Chen et al, 2022; Chen et al, 2014; Filbey et al, 2014; Maizels and Gause, 2023). In future studies, it would be interesting to target TGM4 by active or passive immunisation and monitor the effects on myeloid cell subsets during parasite infection.

An intriguing observation is the ~100-fold reduced affinity of TGM4 for TβRII compared to TGM1. Although we have yet to solve the 3-dimensional structure of TGM4, we note from the sequence alignment (Fig. EV1B) that key residues (K254, N255) of TGM1 are substituted by serine and histidine, respectively, perhaps

compromising the ion bridge to TβRII D55 observed in other members of the gene family (Mukundan et al, 2022). A broader question is what evolutionary advantage may be gained by the reduced affinity of TGM4? One answer may be that this allows the ligand to be more co-receptor-dependent, and thus discriminatory, eliciting cell-specific responses despite interaction of TGM4 with ubiquitously expressed TβRI and TβRII. It is likely that the selectivity of TGM4 is based on a continuum threshold; although MFB-F11 fibroblasts do express the CD44 co-receptor (and can bind TGM4 in flow cytometry assays), either the expression level is too low, and/or the additional co-receptors required for high affinity interactions are absent, resulting in the failure of TGM4 to assemble an activation complex. The ability of TGM1 to stimulate fibroblasts can thus be attributed to its >tenfold higher affinity for TβRII. Thus, the nuances of differential affinities for TβRI and TβRII, and the disparate levels of expression of essential co-receptors in diverse cell types, determine whether receptor ligation is sufficient to drive signalling. It is also possible that the TGM4 co-receptors directly deliver secondary intracellular signals that contribute to differential cell responses of TGM4, a question that we are now investigating.

The affinities of TGM family members for TβRs may thus be calibrated by evolution to depend on interaction with co-receptors such as CD44 that is bound by both TGM1 (van Dinther et al, 2023) and TGM4 interactions. Notably, TGM4 interacts with a broader range of co-receptor partners, including CD49d, CD206 and NRP-1, and while CD44 interactions are governed by D4–5, CD49d interacts with D1-3. This may explain why while cell binding by TGM4 is primarily mediated by D4–5, it is diminished compared to full-length TGM4 in which D1-3 may contribute.

CD49d is an integrin α chain subunit (α4, Itga4), that combines with β integrins; dimerised to β1 integrin (CD29) it comprises the VLA (very late antigen) 4 surface marker that mediates binding to VCAM1 (CD106), expressed on eosinophils, myeloid cells and mesenchymal stem cells, in an interaction controlling leukocyte endothelial adhesion, rolling and extravasation. When combined with β7 integrin, CD49d forms a homing receptor which binds mucosal vascular addressin cell adhesion molecule 1 (MAdCAM) in mucosal tissues. CD49d expression is raised in many stimulated cell types, such as activated eosinophil subsets compared to basal

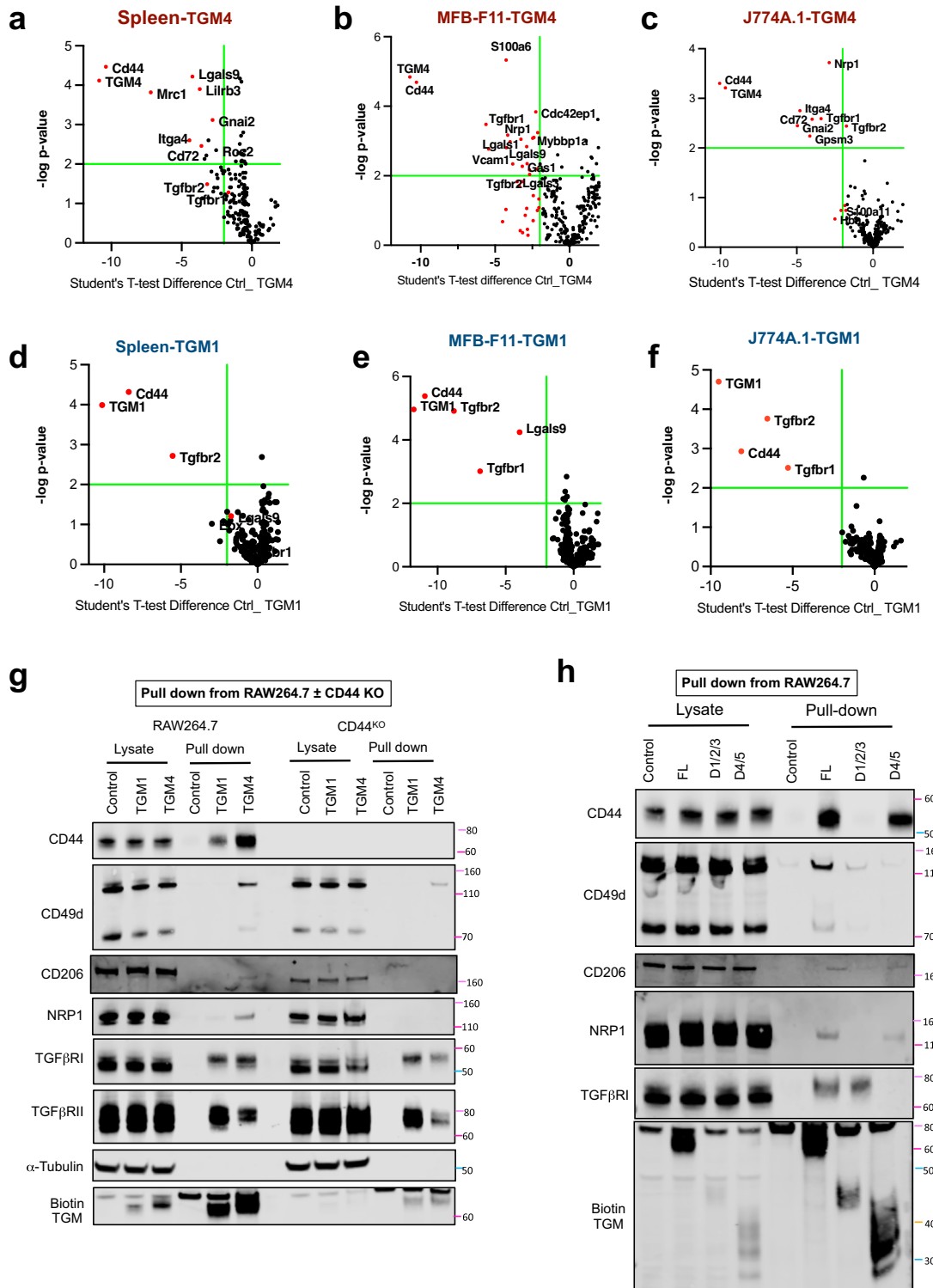

**Figure 5. Identification of novel co-receptors for TGM4.**

(A–C) Pull-down samples purified from cells treated with biotin-tagged TGM4 and precipitated with streptavidin resin were subjected to mass spectrometry and analysed relative to control samples, pooling data from three biological replicate experiments, using C57BL/6 strain murine splenocytes (A), MFB-F11 fibroblasts (B) and J774 macrophages (C). (D–F) Parallel analyses of pull-down samples in the same experiments with TGM1 in the three indicated cell types, pooling data from three biological replicate experiments. (G) Pull-down and western blot analysis in CD44-sufficient and -deficient RAW264.7 cells, probed with antibodies to the indicated proteins. (H) As (G), but comparing full-length and truncated constructs of TGM4 by pulldown and western with antibodies to indicated proteins. Data Information: Data in (A–F) analysed by Student's *t* test plotted on graph by difference against the indicated *P* values plotted as negative logarithmic values.

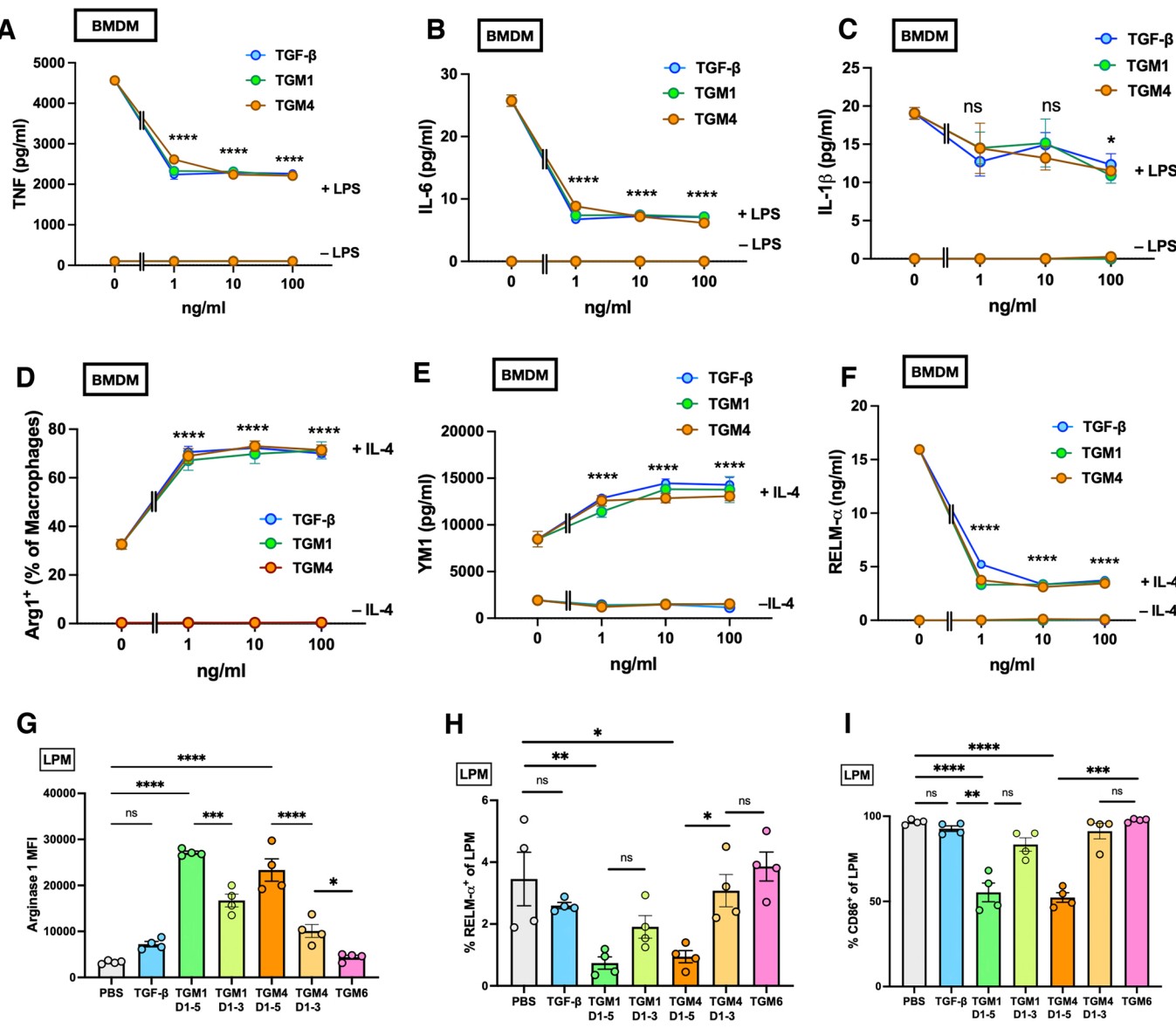

**Figure 6. TGM4 activity on macrophage populations.**

(A, B) In vitro responses of bone marrow-derived macrophages (BMDM) to a range of concentrations of TGF-β, TGM1 and TGM4, in the presence or absence of 100 ng/ml LPS, measured by release of TNF (A) and IL-6 (B) after 24 h of culture. Data represent one of three biological replicate experiments, with $n = 3$, showing mean ± SE. (C) In vitro responses of bone marrow-derived macrophages (BMDM) cells to a range of concentrations of TGF-β, TGM1 and TGM4, in the presence or absence of 100 ng/ml LPS, measured by release of IL-1β after 24 h of culture. Data represent 1 of 2 biological replicate experiments, with $n = 3$, showing mean ± SE. (D–F) Responses of bone marrow-derived macrophages(BMDM) to a range of concentrations of TGF-β, TGM1 and TGM4, in the presence or absence of 20 ng/ml IL-4, measured by release of Arginase-1 (D), Chi3L3 (Ym1) (E), and RELM-α (F) after 24 h of culture. Data represent one of two biological replicate experiments, with $n = 3$, except for (F), which is 1 of 4 replicates, showing mean ± SE. (G–I) Phenotype of resident large peritoneal macrophages(LPM), collected from the peritoneal cavity 24 h after i.p. injection of PBS, TGF-β, TGM1 or TGM4, and analysed by flow cytometry for staining for Arginase-1 (G), RELM-α (H), and CD86 (I) Data represent one of three biological replicate experiments, with $n = 3$, showing mean ± SE. Data Information: Data in (A–F) analysed by two-way ANOVA with Dunnett's multiple comparisons test. In (A, B, D–F), statistical significances for each dose of TGF-β, TGM1 and TGM4 in presence of LPS versus LPS alone was $P < 0.0001$; in (C), only doses of 100 ng/ml reached significance, at $P = 0.0358$ for TGF-β, 0.0101 for TGM1, and 0.0173 for TGM4. Data in (G–I) were analysed by one-way ANOVA; in (G), TGM1 D1-5 vs PBS, and TGM4 D1-5 vs PBS, both $P < 0.0001$; TGM1 D1-5 vs TGM1 D1-3, $P = 0.001$; TGM4 D1-5 vs TGM4 D1-3, $P < 0.0001$; TGM4 D1-3 vs TGM6, $P = 0.0456$. in (H), TGM1 D1-5 vs PBS, $P = 0.0062$, and TGM4 D1-5 vs PBS, $P = 0.0127$; TGM4 D1-5 vs TGM4 D1-3, $P = 0.0451$. In (I), TGM1 D1-5 vs PBS, and TGM4 D1-5 vs PBS, both $P < 0.0001$; TGM1 D1-5 vs TGF-β, $P = 0.0038$; and TGM4 D1-5 vs TGM6, $P = 0.0004$. For all panels, ns = not significant, $*P < 0.05$, $**P < 0.01$, $***P < 0.001$, $****P < 0.0001$.

populations (Gurtner et al, 2023). Thus, by targeting CD49d, TGM4 can encompass multiple immune cell subsets that are instrumental to protective immunity against infection.

Other integrins facilitate the activation of latent TGF-β (Travis and Sheppard, 2014; Worthington et al, 2011); for example, αv combined with different β subunits, releasing mature TGF-β from the latency-associated peptide within the extracellular matrix (Shi et al, 2011; Wang et al, 2017); however α4 integrin is not implicated in this process (Hinz, 2015; Nolte and Margadant, 2020). As we ascertained that a neutralising anti-TGF-β antibody did not reduce signal from TGM4, it is unlikely that release of active TGF-β from the host contributes significantly to the effect observed by TGM4.

Future work will aim not only to provide finer detail on how TGM proteins interface with multiple receptors, at the structural level and in terms of target cell populations, but also to evaluate the in vivo role of these products. Although it is not yet possible to gene target *H. polygyrus* and similar nematode parasites, antibody neutralisation experiments are a feasible approach to test if TGM proteins are essential for helminth survival in the host. Furthermore, TGM1 has proven to be effective in multiple mouse models of inflammation (Chauché et al, 2022; Redgrave et al, 2023; Smyth et al, 2021; Smyth et al, 2023); the therapeutic potential of TGM4 may be further enhanced in settings where selective modulation of the myeloid cell population while minimising fibrotic responses is the most desirable outcome.

# Methods

### Reagents and tools table

| Reagent/resource | Reference or source | Identifier or catalogue number |
|---|---|---|
| **Reagents** | | |
| GFP-TRAP beads | Chromotek | GTA-20 |
| Halt™ Phosphatase Inhibitor Cocktail | Thermo Scientific™ | 78427 |
| Halt™ Protease Inhibitor Cocktail (100X) | Thermo Scientific™ | 78438 |
| EZ-Link™ Sulfo-NHS-LC-Biotin, No-Weigh™ Format | Thermo Scientific™ | A39257 |
| Pierce™ NeutrAvidin™ Agarose | Thermo Scientific™ | 29201 |
| Cell Lysis Buffer (10X) | CST | 9803 |
| Lipofectamine™ 2000 Transfection Reagent | Invitrogen™ | 11668500 |
| Lipofectamine™ 3000 Transfection Reagent | Invitrogen™ | L3000150 |
| Lipofectamine™ LTX Reagent | Invitrogen™ | 15338500 |
| Alexa Fluor™ 488 Microscale Protein Labeling Kit | Invitrogen™ | A30006 |
| Alexa Fluor™ 594 Microscale Protein Labeling Kit | Invitrogen™ | A30008 |
| DTT | Melford | 3483-12-3 |
| Precision Red Advanced Protein Assay: 1× stock | Cytoskeleton | ADV02 |
| PD MiniTrap desalting columns with Sephadex G-25 resin | Cytiva | 28918007 |

| Reagent/resource | Reference or source | Identifier or catalogue number |
|---|---|---|
| One Shot™ Stbl3™ Chemically Competent *E. coli* | Invitrogen™ | C737303 |
| VeriFi™ Hot Start Mix | PCR Biosystems | PB10.46-01 |
| T4 Ligase | NEB | |
| T4 Polynucleotide Kinase | NEB | M0201S |
| Adenosine 5′-Triphosphate (ATP) | NEB | P0756S |
| PstI | NEB | R0140S |
| XhoI | NEB | R0146S |
| AscI | NEB | R0558S |
| NotI | NEB | R0189S |
| FastDigest BpiI | Thermo Scientific™ | FD1014 |
| Tango Buffer (10X) | Thermo Scientific™ | BY5 |
| Novex™ Sharp Pre-stained Protein Standard | Invitrogen™ | LC5800 |
| GeneRuler 1 kb DNA Ladder, ready-to-use | Thermo Scientific™ | SM0313 |
| Pierce™ ECL Plus Western Blotting Substrate | Thermo Scientific™ | 32132 |
| CM-5 SPR sensor chip | Cytiva | BR100012 |
| Neutravidin | Thermo Scientific™ | 31000 |
| **Cell lines** | | |
| MFB-F11 | Tesseur et al, 2006 | 31000 |
| RAW264.7 | | TIB71 |
| J774A.1 | | TIB67 |
| EL4 | | TIB39 |
| HepG2 | Kind Gift from CRUK Beatson Institute | HB8065 |
| CD44$^{KO}$ in MFB-F11 | In this study | |
| CD44$^{KO}$ in RAW264.7 | In this study | |
| CD49d$^{KO}$ in RAW264.7 | In this study | |
| NRP-1$^{KO}$ in RAW264.7 | In this study | |
| expi293 | Invitrogen | A14527 |
| BL21(DE3) | EMD-Millipore | 69450 |
| **Primary antibodies** | | |
| Recombinant Anti-TGF-β Receptor I antibody | Abcam | ab235578 |
| Recombinant Anti-TGF-β Receptor II antibody | Abcam | ab25936 |
| Recombinant Anti-CD44 antibody | | ab189524 |
| Recombinant Anti-CD44 antibody | | ab243894 |
| CD49d (Integrin α 4) Recombinant Rabbit Monoclonal Antibody | Thermo fischer scientific | MA5-35435 |
| CD206/MRC1 (E6T5J) XP® Rabbit mAb | CST | #24595 |
| ChromoTek GFP Monoclonal antibody | Chromotek | 3H9 |
| Anti-CD72 antibody | Abcam | Ab201079 |
| SMAD2/3 Antibody #3102 | CST | #3102, #5678 |

| Reagent/resource | Reference or source | Identifier or catalogue number |
| --- | --- | --- |
| Phospho-SMAD2 (Ser465/467)/ SMAD3 (Ser423/425) (D27F4) Rabbit mAb | CST | #8828 |
| LILRB3 Polyclonal Antibody | | PA590933 |
| Neuropilin-1 | Abcam | ab81321 |
| Anti·His HRP Conjugate Kits | Qiagen | 1014992 |
| Anti-α-Tubulin antibody (DM1A) | Abcam | ab7291 |
| **Secondary antibodies** | | |
| Goat anti-Mouse IgG (H + L) Cross-Adsorbed Secondary Antibody, DyLight™ 680 | Invitrogen | 35519 |
| Goat anti-Rabbit IgG (H + L) Secondary Antibody, DyLight™ 800 4X PEG | Invitrogen | SA535571 |
| Donkey anti-Goat IgG (H + L) Cross-Adsorbed Secondary Antibody, DyLight™ 680 | Invitrogen | SA510090 |
| Goat anti-Rat IgG (H + L) Cross-Adsorbed Secondary Antibody, DyLight™ 680 | Invitrogen | SA510022 |
| Streptavidin Alexa 680 conjugate | Invitrogen | S32358 |
| **Flow cytometry antibodies** | | |
| FITC anti-mouse/human CD44 Antibody | Biolegend | 103006 |
| PE anti-mouse/human CD44 Antibody | Biolegend | 103008 |
| FITC anti-mouse CD4 Antibody | Biolegend | 100406 |
| PE/Dazzle™ 594 anti-mouse CD45 Antibody | Biolegend | 103146 |
| IgG from rat serum | Sigma | I4131 |
| UltraComp eBeads™ Compensation Beads | Invitrogen™ | 01-2222-42 |

## General materials

Details of enzymes, chemicals, substrates, transfection and labelling reagents and other materials purchased from commercial suppliers is given in "Reagents and Tools Table".

## Expression of TGM1 and TGM4 recombinant proteins

For cellular and in vivo applications using live cells, recombinant proteins were expressed in human HEK293 cells. Mammalian codon-optimised genes were synthesised by GeneArt as previously published (Smyth et al, 2018), and subcloned into the mammalian expression vector pSecTag2A using restriction sites *Asc*I and *Not*I or *Asc*I and *Apa*I. Amplification and cloning of the truncated versions of TGM4 (D1-3 and D4–5) was performed by PCR amplification using proofreading Taq polymerase Phusion Hi as per the manufacturer's instructions (Invitrogen), full-length codon-optimised TGM4 as a template, and the primers shown in Appendix Table S3 TGMs were expressed in expi293 cells (Thermo) and purified from the conditioned medium by capturing

on nickel-loaded HiTrap chelating columns (Cytiva). After washing until the UV absorbance returned to baseline, the proteins were eluted with a 0.0–0.5 M imidazole gradient. The fractions with TGM were pooled, concentrated, and further purified on a Superdex 200 16/60 column (Cytiva).

For biophysical procedures, domains D1-2 and D3 were expressed in *E. coli*. DNA inserts coding for TGM4 D2 and TGM4 D3 were inserted into a modified form pET32a (EMD-Millipore) between a *Kpn*I restriction site in the frame and immediately following the thrombin cleavage site and a *Hind*III restriction site in the 3' polylinker. Constructs were expressed in BL21(DE3) cells (EMD-Millipore) cultured at 37 °C to an OD of 0.8, at which point protein expression was induced by adding 0.8 mM isopropyl β-D-1-thiogalactopyranoside (IPTG). Isolation of inclusion bodies, refolding, and final high-resolution ion-exchange purification was performed as described for TGM1-D3 (Mukundan et al, 2022).

## Production and purification of TGF-β family and CD44 receptor extracellular domains (ECDs)

Biotinylated avi-tagged human TβRI and TβRII extracellular domains (ECDs) were produced as insoluble proteins in *E. coli*, and after reconstitution, refolding, and purification, were enzymatically modified with purified recombinant BirA, as previously described (Mukundan et al, 2022). The human ActRII and BMPRII receptor ECDs, preceded by the rat serum albumin signal peptide, a hexahistidine tag and a thrombin cleavage site, were inserted into a pcDNA 3.1+ vector between the *Nhe*I and *Xho*I sites. The constructs were expressed in expi293 cells (Thermo) and purified from the conditioned medium in an identical manner to that described above for the TGMs. The mouse and human CD44 ECDs were expressed in expi293 cells and purified from the conditioned medium as previously described (van Dinther et al, 2023).

## Labelling of TGM1/TGM4 for use in flow cytometry

Recombinant TGM1 and TGM4 D1-5, D1-3 and D4–5 proteins were fluorescently labelled with Alexa Fluor™ 594 or 488 Microscale Protein Labelling Kits (Invitrogen™, A30008 or A30006) as described (van Dinther et al, 2023). Briefly, 50 µg (~1 mg/ml) protein was mixed with Alexa dye and 1 M sodium bicarbonate at 1/10th of the reaction volume concentration, and incubated at room temperature for 15 min. Unlabelled dye was removed from the reaction mixture on a desalting column supplied in the labelling kit. Protein concentrations were calculated using a Nanodrop spectrophotometer (Thermo Scientific). The degree of labelling (DOL, the average number of dye molecules per protein molecule) of the dye-conjugated TGMs was 7.8, 3.6 and 6.8 for TGM1-AF488, TGM1-AF594 and TGM4-AF488, respectively.

## Cell lines

All reagents and cell lines used in this study are listed in "Reagents and Tools Table". HepG2 cells were a kind gift from Dr. Saverio Tardito, CRUK Beatson Institute, Glasgow. All cell lines used in this study were grown and maintained in DMEM supplemented with 10% foetal bovine serum, 2 mM L-glutamine and 100 I.U./ml penicillin–streptomycin in tissue culture dishes or flasks at 37 °C, 5% $CO_2$.

## Primary splenocytes

Spleens recovered from C57BL/6J mice were pressed through a 100-μm strainer, flushed with 20 ml RPMI1640 medium to obtain single-cell suspensions. Cells were pelleted at $400 \times g$, and resuspended in 1 ml red blood cell lysis buffer (Sigma) for 5 min at room temperature. Cells were washed and resuspended in RPMI1640 medium, and counted using a haemocytometer in diluted trypan blue solution.

## Bone marrow-derived macrophages

Bone marrow was recovered from the femurs and tibias of naive C57BL/6J or BALB/c mice. Briefly, the connective and muscular tissue were removed, bones washed, and the tips of the epiphyses cut using a sterile scissors and forceps. The bone marrow was then flushed out with DMEM using a 25-gauge needle into a petri dish. The flecks of bone marrow were then homogenised using a 19-gauge needle before filtration through a 70-μm filter into a sterile tube. The single-cell suspension was then centrifuged at $400 \times g$ for 5 min at 4 °C. Cells were then counted and resuspended at $1 \times 10^6$ cells/ml in DMEM supplemented with 30% L929 media containing CSF-1. Cells were then incubated for 6–7 days, with fresh media added on day 3. On the final day, cells were harvested by washing with PBS to remove any potential non-adherent cells, followed by scraping to remove the adherent macrophages. BMDM were then counted, resuspended at the appropriate concentration and plated in 96-well plates for ELISA experiments, or 24-well plates for flow cytometric experiments. Cells were allowed to rest for 1–2 h before stimulation to allow for adherence to the new plate.

## Reporter bioassays

The TGF-β bioassay with MFB-F11 cells developed by Tesseur et al (Tesseur et al, 2006) was performed as previously described (Johnston et al, 2017). MFB-F11 cells were tested and found to be mycoplasma-free. Briefly, confluent cells were detached with trypsin, and resuspended in DMEM with 2.5% FCS, 100 U/ml of penicillin, 100 μg/ml of streptomycin and 2 mM L-glutamine at a concentration of $8 \times 10^5$ cells/ml. In 50 μl, $4 \times 10^4$ cells were added to each well of a 96-well round-bottomed plate. Dilutions of purified proteins were then added to each well in a volume of up to 50 μl and incubated for 24 h at 37 °C. Subsequently, 20 μl of supernatant were aspirated from each well, added to an ELISA plate (Nalge Nunc International, USA) with 180 μl of reconstituted Sigma FastTM p-nitrophenyl phosphate substrate and incubated at room temperature in the dark for up to 4 h. Plates were read on at 405 nm on an Emax precision microplate reader (Molecular Devices, USA). All conditions were set up in duplicate and repeated at least twice.

## Western blotting

Cell lysates and pull-down samples were analysed on 4–12% Bis-Tris SDS-PAGE gels and transferred onto nitrocellulose membrane using iBlot2 (Invitrogen, IB21001). Membranes were treated in 5% non-fat milk blocking solution for 1 h and incubated with primary antibodies listed in the Reagents and Tools Table (1:1000 in 5% BSA containing TBST) overnight at 4 °C and washed 3× (5 min) using 1× TBST. Fluorescently conjugated secondary antibodies as listed in the Reagents and Tools Table, diluted 1:10,000 in 5% BSA containing TBST were used to detect the protein bands by Odyssey CLx Imaging System (LI-COR Biosciences).

## Foxp3⁺ Treg induction assay

A single-cell suspension was prepared from the spleens of naive BALB/c or Foxp3-GFP BALB/c transgenic mice (Fontenot et al, 2005), with 2 min incubation in red blood cell lysis buffer (Sigma). Cells were then washed and resuspended in DMEM containing HEPES (Gibco), supplemented with 2 mM L-glutamine, 100 U/ml of penicillin and 100 μg/ml of streptomycin (Gibco), 10% heat-inactivated FCS (Gibco), and 50 nM 2-mercaptoethanol (Gibco). Naive CD4⁺ T cells were isolated by magnetic sorting using the mouse naive CD4⁺ T-cell isolation kit on the AutoMACS system (Miltenyi, Germany) as per the manufacturer's instructions. Cells were cultured at $2 \times 10^5$ per well in flat-bottomed 96-well plates (Corning, USA) with the addition of IL-2 (Miltenyi) at a final concentration of 400 U/ml and pre-coated with 10 μg/ml of anti-CD3 (eBioscience). Cells were cultured at 37 °C in 5% CO₂ for at least 72 h before being removed for flow cytometric analysis. For TβRI (ALK5) inhibitor assays, 5 μM SB431542 (Tocris Bioscience, UK) was added, with DMSO added to control wells.

## Surface staining with antibodies and labelled TGM

Cells were prepared for flow cytometric analysis in PBS, incubated with Fixable Viability Dye eFluor™ 506 at a dilution of 1:1000 in the dark for 25 min at 4 °C. Thereafter, cells were washed twice in FACS buffer. In some assays, LIVE/DEAD® fixable blue (Life Technologies, USA) was diluted to 1:1000 in PBS; 100 μl were added to each sample of cells, which was then incubated in the dark for 20 min at 4 °C and washed twice in FACS buffer (1× PBS, 0.5% (w/v) BSA, 0.05% sodium azide). Fc receptors were blocked by incubating cells with 1:50 anti-mouse CD16/CD32 (Fc block, Invitrogen) for 10 min at 4 °C, followed by two washes with FACS buffer. Antibodies used are listed in the Reagents and Tools Table. Separate Foxp3 staining was not required as cells were from Foxp3-GFP transgenic mice. Labelled TGMs were serially diluted in brilliant stain buffer and cells incubated for 20 min at 4 °C. Cells were washed twice with FACS buffer and filtered before acquisition on a BD FACSCelesta cytometer. Wherever available, isotype controls and fluorescence minus one (FMO) controls were used. Single-stained compensation beads were used for compensation settings. FACS data was analysed using FlowJo and Prism GraphPad software.

## ImageStream analysis

Bone marrow macrophages were isolated after 6–7 days of differentiation with L929 media and placed in serum-free DMEM overnight in 24-well plates. The cells were stimulated with 20 ng/ml TGF-β, TGM1 or TGM4 in serum-free media for 1 hr, after which the samples were fixed with 200 ml of 4% paraformaldehyde (PFA) solution in PBS (Sigma) to reach a final concentration of 2%. The cells were left to fix in a 37 °C water bath for 20 min followed by centrifugation at $600 \times g$ for 6 min, and subsequently washed twice in PBS. After the final wash cells were permeabilised in perm-wash solution (1× PBS with 0.1%. Triton X-100) in the presence of 0.2 mg/ml of anti-SMAD2/3 AF488 antibody (Santa Cruz Biotechnology, Dallas, TX, USA) and 0.2 mg/ml rat IgG (Sigma). The

samples were incubated in the fridge in the dark for 40 min and after were washed twice in 1 ml of FACS buffer, using $600 \times g$ centrifugation, and finally resuspended in 0.5 mM DRAQ5 (BioLegend) in PBS. The samples were analysed on the Amnis ImageStream X MKII and were processed using the IDEAS software (Amnis, Seattle, WA, USA). Following this, cells were selectively gated on the focused cells, then single cells, and those exhibiting double positivity for SMAD2/3-AF488 and DRAQ5. The determination of nuclear localisation was achieved by computing the similarity dilate score between the two markers. A higher score signified nuclear localisation of SMAD2/3.

## Surface plasmon resonance

All SPR experiments were performed with a BIAcore X100 system (Cytiva). Neutravidin was coupled to the surface of a CM-5 chip (Cytiva) by EDC-NHS activation of the chip, followed by injection of neutravidin (Thermo) over the surface in sodium acetate, pH 4.5 until the RU increased by 6000–15,000 RU. Biotinylated Avi-tagged TβRI and biotinylated Avi-tagged TβRII were captured onto the chip surface at a maximum density of 150 RU. All experiments were performed in HBS-EP buffer, 25 mM HEPES, 150 mM NaCl, 0.005% P20 surfactant, pH 7.4, at an injection rate of $100 \, \mu L \, min^{-1}$. The surface was regenerated in between each injection with a 30 s injection of 1 M guanidine hydrochloride. The experimental sensorgrams were obtained with double referencing with a control cell coated similarly with neutravidin but lacking the captured receptor and 8 blank buffer injections at the beginning of the run before injection of the samples. The data was analysed by fitting the results to a 1:1 kinetic model using the SPR analysis software Scrubber (BioLogic Software).

## Isothermal titration calorimetry

All ITC experiments were performed with a Microcal PEAQ-ITC system (Malvern Instruments). All experiments were performed in 25 mM $Na_2HPO_4$, 50 mM NaCl, pH 6.0 at 35 °C, with 15 2.5 μL injections with a duration of 5 s, a spacing of 150 s, and a reference power of 10. All samples were dialysed against the same ITC buffer before loading into the system. Two independent datasets were globally fit using the programs NITPIC (Keller et al, 2012), SEDPHAT (Brautigam et al, 2016; Zhao et al, 2015) and GUSSI (Brautigam, 2015).

## Nuclear magnetic resonance

NMR samples were prepared in 25 mM $Na_2HPO_4$, 50 mM NaCl, pH 6.0 and were transferred to 5 mm susceptibility-matched NMR microtubes for data collection. NMR data was collected with Bruker 600, 700 or 800 MHz spectrometers equipped with a 5 mm $^1H$ ($^{13}C$, $^{15}N$) z-gradient "TCI" cryogenically cooled probe at a temperature of 37 °C. $^1H$-$^{15}N$ HSQC spectra were acquired as described, with water flipback (Grzesiek and Bax, 1993) and WATERGATE suppression pulses (Piotto et al, 1992). NMR spectra were processed, analysed, and visualised using NMRPipe (Delaglio et al, 1995) and NMRFAM-SPARKY (Lee et al, 2015).

## Cell transfection

Extrachromosomal expression plasmids were transfected into MFB-F11 cells with Lipofectamine-2000, RAW264.7 cells with

Lipofectamine LTx and HepG2 with Lipofectamine 3000 according to the manufacturer's instruction. In six-well plates, $3 \times 10^5$ cells were allowed to adhere overnight. In total, 6 μl lipofectamine reagent and 2.5 μg plasmid DNA diluted and mixed in 200 μl serum-free DMEM in two separate tubes. Later, diluted plasmid DNA and lipofectamine were pooled together and mixed well and incubated at room temperature (10 min). This mixture was added to the cells and mixed by swirling the plate. Transfection to cells to be used for GFP-TRAP pulldown were performed in 15-cm tissue culture plates.

## Molecular cloning and CRISPR-Cas9 knockout

To engineer pSecTag2A-eGFP plasmid, eGFP was PCR amplified from pEGFP-N1 using primers described in Appendix Table S3 and inserted in pSecTag2A plasmid using *PstI* and *XhoI*. TGM1 D1-5, TGM4 D1-5, D1-3 and D4–5 coding sequences were PCR amplified using primers mentioned in Appendix Table S3 and were inserted using *AscI* and *NotI* into the pSectAg2A-eGFP plasmid to generate pSecTag2A-TGM-eGFP constructs.

For deletion of CD44, CD49d and NRP-1 expression in RAW264.7 macrophage or MFB-F11 fibroblast cell lines, a CRISPR strategy was used (Ran et al, 2013), with guides mentioned in Appendix Table S4 cloned in pSpCas9(BB)-2A-GFP (a kind gift from Dr. Jamie Whitelaw, CRUK Beatson institute, Glasgow; Addgene plasmid #48138). Overall, 2.5 mg empty and CD44 guide RNA containing pSpCas9(BB)-2A-GFP plasmids were transfected in MFB-F11 or RAW264.7 cells. Twenty-four hours post transfection, GFP-positive single cells were sorted by FACS in 96-well plates. Single-cell clones were screened by western blotting after approximately 2 weeks.

## GFP-TRAP pulldown

For expression of eGFP fusion proteins, MFB-F11, RAW and HepG2 cells were grown to 70% confluence in 150 mm Petri dishes. Approximately 50 μg of pSecTag2A-eGFP, or the pSecTag2A-eGFP plasmid containing in-frame fusions with TGM1 and TGM4 full-length (FL), D1-3 or D4–5 were transfected into MFB-F11 or RAW or HEPG2. Forty hours post transfection, cells were lysed with cell lysis buffer (10 mM Tris-HCl pH 7.5, 150 mM NaCl, 0.5 mM EDTA and 0.5% NP-40) supplemented with 1× Halt protease and phosphatase inhibitor cocktails (Invitrogen). Cell lysates were cleared by centrifugation ($13,000 \times g$, 10 min). In total, 2 mg of cell lysates were incubated with 25 ml of GFP-TRAP beads (Chromotek, GTA-20) for 1 h at 4 °C on rotation. Beads were washed 4× with cell lysis buffer (5 min each wash on rotation). To elute the proteins from the beads, 50 μL 2× NuPAGE LDS sample buffer with 25 mM DTT was added and boiled (100 °C, 5 min). Protein samples were analysed on 4–12% Bis-tris acrylamide gels followed by western blotting.

## Streptavidin pulldown

Ten μg of TGM1 and TGM4 D1-5, D1-3 and D4–5 were biotinylated and purified as described (van Dinther et al, 2023). For the pulldown, MFB-F11, RAW264.7 and CD44 knockout cells were grown at 80–90% confluency in 15-cm tissue culture dishes, washed 3× with ice-cold PBS and incubated with ~3.5 μg of biotinylated TGMs for 3 h on ice. Cells were washed 3× with ice-

cold PBS and lysed with Cell Lysis Buffer (100 mM NaCl, 25 mM Tris, pH 7.5, 5 mM MgCl$_2$ and 0.5% NP-40) supplemented with 1× Halt protease inhibitor (Thermo Scientific, 1861279) and phosphatase inhibitor (Thermo Scientific, 78427) cocktails. Cell lysates were cleared by centrifugation (13,000 × $g$, 10 min). Two µg of cell lysates were incubated with 30 µl of Neutravidin agarose beads (Thermo Scientific, 29201) for 1 h at 4 °C. Beads were washed with lysis buffer 4× (5 min each). For mass spectrometry, beads were stored in 100 mM ammonium bicarbonate at −20 °C. For western blotting, 50 µl LDS sample buffer (Invitrogen, NP0007) containing 25 mM dithiothreitol (DTT) was added to beads and heated for 5 min at 100 °C.

### Liquid chromatography and mass spectrometry (LC-MS)

Neutravidin agarose beads were resuspended in a 2 M Urea and 100 mM ammonium bicarbonate buffer and stored at −20 °C. Three biological replicates for each condition were digested with Lys-C (Alpha Laboratories) and trypsin (Promega) "on beads" as previously described (Hubner et al, 2010). Peptides resulting from all trypsin digestions were separated by nanoscale C18 reverse-phase liquid chromatography using an EASY-nLC II 1200 (Thermo Scientific) coupled to an Orbitrap Q-Exactive HF mass spectrometer (Thermo Scientific). Elution was carried out at a flow rate of 300 nl/min using a binary gradient, into a 20 cm fused silica emitter (New Objective) packed in-house with ReproSil-Pur C18-AQ, 1.9 µm resin (Dr Maisch GmbH), for a total run-time duration of 125 min. Packed emitter was kept at 35 °C by means of a column oven (Sonation) integrated into the nanoelectrospray ion source (Thermo Scientific). Eluting peptides were electrosprayed into the mass spectrometer using a nanoelectrospray ion source. An Active Background Ion Reduction Device (ESI Source Solutions) was used to decrease air contaminants signal level. The Xcalibur 4.2 software (Thermo Scientific) was used for data acquisition. A full scan was acquired at a resolution of 120,000 at 200 $m/z$, over mass range of 350–1400 $m/z$. HCD fragmentation was triggered for the top 15 most intense ions detected in the full scan. Ions were isolated for fragmentation with a target of 1E5 ions, for a maximum of 125 ms, at a resolution of 15,000 at 200 $m/z$. Ions that have already been selected for MS2 were dynamically excluded for 20 s.

### MS data analysis

The MS Raw data were processed with MaxQuant software (Cox and Mann, 2008) version 1.6.14.0 and searched with Andromeda search engine (Cox et al, 2011), querying SwissProt (UniProt C, 2019) *Mus musculus* (25198 entries). First and main searches were performed with precursor mass tolerances of 20 ppm and 4.5 ppm, respectively, and MS/MS tolerance of 20 ppm. The minimum peptide length was set to six amino acids and specificity for trypsin cleavage was required. Cysteine carbamidomethylation was set as fixed modification, whereas Methionine oxidation, Phosphorylation on Serine-Threonine-Tyrosine, and N-terminal acetylation were specified as variable modifications. The peptide, protein, and site false discovery rate (FDR) was set to 1%. All MaxQuant outputs were analysed with Perseus software version 1.6.13.0 (Tyanova et al, 2016). The raw files and the MaxQuant search results files have been deposited as a complete submission to the Proteome Xchange

Consortium (Deutsch et al, 2023) via the PRIDE partner repository (Perez-Riverol et al, 2022) with the dataset identifier PXD048278.

Protein abundance was measured using label-free quantification (LFQ) intensities reported in the ProteinGroups.txt file. Only proteins quantified in all replicates in at least one group, were measured according to the label-free quantification algorithm available in MaxQuant (Cox et al, 2014). Missing values were imputed separately for each column, and significantly enriched proteins were selected using a permutation-based Student's $t$ test with FDR set at 5%.

### Animal use and ethics approval

Animal experiments and maintenance of the Foxp3-GFP transgenic mouse line were conducted under a UK Home Office licence and approved by the University of Glasgow Animal Welfare and Ethical Review Board. All mice were maintained on standard diet in individually ventilated cages within a barrier facility. Sample size groups of 5 mice were used, selecting randomly from a single batch, and no animals were excluded from the analysis. As data were collected objectively by flow cytometry, no blinding measures were undertaken.

### Quantification, statistical analysis and software

Western blotting was quantified using ImageJ (FIJI). FACS data were analysed with FlowJo. Unpaired Sstudent's $t$ test or ANOVA were performed using GraphPad Prism on the basis that data were normally distributed. A full list of software used in this study is presented in Appendix Table S5.

## Data availability

The source data for this study is deposited online with Biostudies (S-BSST1623). The raw mass spectrometry files and the MaxQuant search results files have been deposited as a complete submission to the Proteome Xchange Consortium (Deutsch et al, 2023) via the PRoteomics IDEntification (PRIDE) database repository (Perez-Riverol et al, 2022) with the dataset identifier PXD048278: https://www.ebi.ac.uk/pride/archive/projects/PXD048278/.

The source data of this paper are collected in the following database record: biostudies:S-SCDT-10_1038-S44319-024-00323-2.

## Peer review information

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

## Acknowledgements

SPS, DJS, KTC, MPJW, TC and CC were supported by the Wellcome Trust through an Investigator Award to RMM (Ref 219530), and the Wellcome Trust core-funded Wellcome Centre for Integrative Parasitology (Ref: 104111); AS was supported by the Lung Foundation Netherlands (project AWWA). AM, C-HB, CSH and APH by NIH grants R03 AI153915 and F30 AI157069; SL, SZ, CS and GI by the Cancer Research UK Beatson Institute; and MvD and PtD by Oncode Institute base funds. The authors gratefully acknowledge assistance and expertise from the Flow Core Facility and the Wolfson Research Facility at the University of Glasgow. For the purpose of open access, the author(s) has applied a Creative Commons Attribution (CC BY) licence to any Author Accepted Manuscript version arising from this submission.

## Author contributions

**Shashi P Singh**: Conceptualisation; Resources; Formal analysis; Validation; Investigation; Methodology; Writing—review and editing. **Danielle J Smyth**: Conceptualisation; Resources; Investigation; Methodology; Writing—review and editing. **Kyle T Cunningham**: Conceptualisation; Investigation; Methodology; Writing—original draft. **Ananya Mukundan**: Formal analysis; Investigation; Writing—review and editing. **Chang-Hyeock Byeon**: Investigation. **Cynthia S Hinck**: Investigation; Methodology. **Madeleine P J White**: Investigation; Methodology. **Claire Ciancia**: Supervision; Investigation. **Natalia Wąsowska**: Investigation. **Anna Sanders**: Investigation. **Regina Jin**: Investigation. **Ruby F White**: Investigation. **Sergio Lilla**: Data curation; Formal analysis; Investigation. **Sara Zanivan**: Supervision; Methodology. **Christina Schoenherr**: Investigation. **Gareth J Inman**: Supervision; Investigation; Methodology; Writing—review and editing. **Maarten van Dinther**: Conceptualisation; Validation; Investigation; Methodology; Writing—review and editing. **Peter ten Dijke**: Conceptualisation; Supervision; Funding acquisition; Investigation; Methodology; Project administration; Writing—review and editing. **Andrew P Hinck**: Conceptualisation; Resources; Formal analysis; Supervision; Funding acquisition; Investigation; Methodology; Project administration; Writing—review and editing. **Rick M Maizels**: Conceptualisation; Resources; Supervision; Funding acquisition; Investigation; Writing—original draft; Project administration; Writing—review and editing.

Source data underlying figure panels in this paper may have individual authorship assigned. Where available, figure panel/source data authorship is listed in the following database record: biostudies:S-SCDT-10_1038-S44319-024-00323-2.

## Disclosure and competing interests statement

The authors declare no competing interests.

# Expanded View Figures

**A**

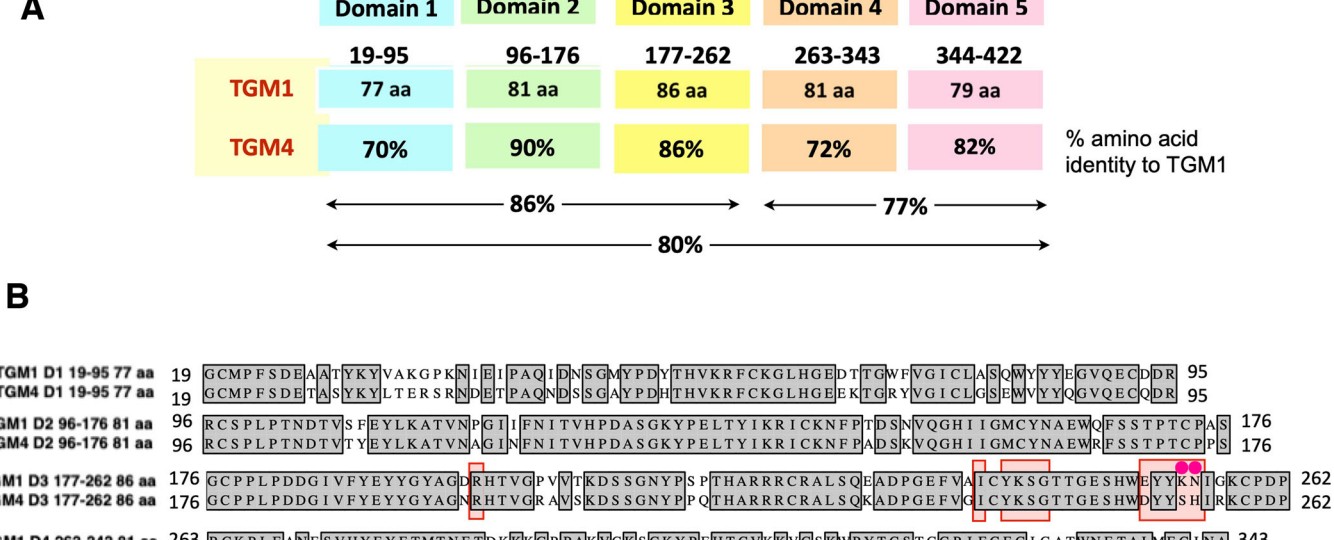

**B**

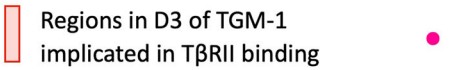

**C**

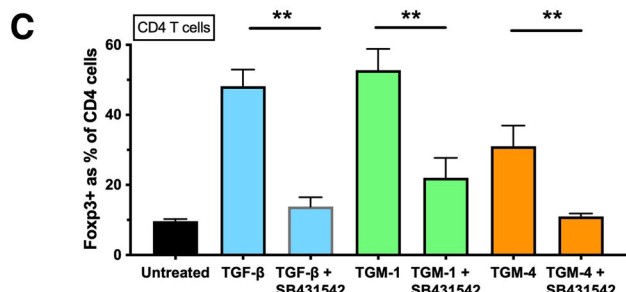

**Figure EV1. Schematic structure and similarity of TGM1 and TGM4 domains.**

(A) Schematic of domain organisation; figures denote amino acid identity for each domain to TGM1, and amino acid identity for D1-3, D4–5 and full-length TGM4. (B) Amino acid alignments for each domain of TGM1 and TGM4; identical domains are shaded. Red background denotes residues of TGM1 identified as contacting TβRII (Mukundan et al, 2022). (C) Inhibition of Foxp3 induction of TGM1 and TGM4 in the presence of SB431542, which blocks kinase activity of ALK5, receptor I for TGF-β; data represent a single experiment, $n = 3$ per group, showing mean ± SE. Data Information: Data in (C) analysed by unpaired $t$ test; untreated and SB431542 treated comparisons, for TGF-β, $P = 0.0040$; for TGM1, $P = 0.0031$; for TGM4, $P = 0.0043$. **$P < 0.01$.

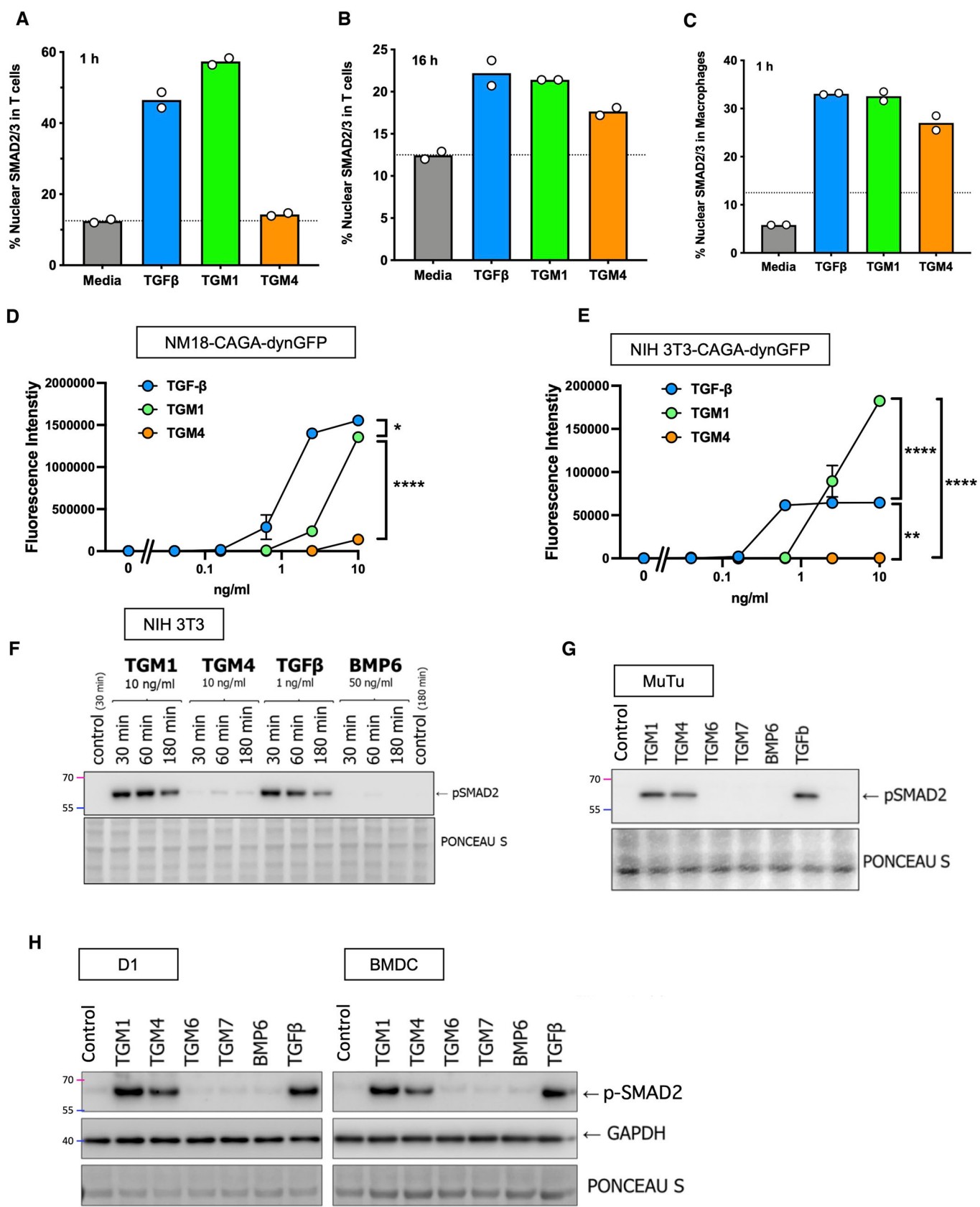

**Figure EV2.   Responses of different cell types to TGM4.**

(A–C) SMAD2/3 nuclear localisation by imaging flow cytometry in T cells at 1 h (A) and 16 h (B) post-stimulation, and in macrophages at 1 h (C), stimulated with TGF-β, TGM1 or TGM4, evaluated by ImageStream. Data are from one ($n = 2$) of two biological replicate experiments. (D, E) NM18 mouse mammary gland epithelial cell (D) and NIH 3T3 mouse embryonic fibroblast (E) lines transfected with the CAGA-dynGFP reporter plasmid, and stimulated with TGFβ, TGM1 or TGM4, assayed by fluorescent intensity at 24 h. Data are from one ($n = 3$) of two biological replicate experiments, presented as mean ± SE. (F) NIH 3T3 cells analysed for pSMAD induction by Western blot by the indicated concentrations of TGF-β, TGM1 or TGM4, for 30, 60 or 180 min. (G) pSMAD induction in MuTu mouse splenic dendritic cells. Cells were stimulated for one hour with 10 ng/ml of each TGM protein, 50 ng/ml BMP6 and 5 ng/ml TGF-β. (H) pSMAD induction in the D1 mouse dendritic cell line (left) and bone marrow-derived DCs, differentiated in vitro with GM-CSF (right). Cells were stimulated for one hour with 10 ng/ml of each TGM protein, 50 ng/ml BMP6 and 5 ng/ml TGF-β. Data Information: Data in (D, E) analysed by two-way ANOVA with Tukey's multiple comparisons test; in (D), at 10 ng/ml, TGF-β vs TGM1 $P = 0.0295$, and both TGF-β or TGM1 vs TGM4 $P < 0.0001$. In (E), TGM1 versus either TGF-β or TGM4, $P < 0.0001$; TGF-β vs TGM4, $P = 0.0053$. *$P < 0.05$, **$P < 0.01$, ****$P < 0.0001$.

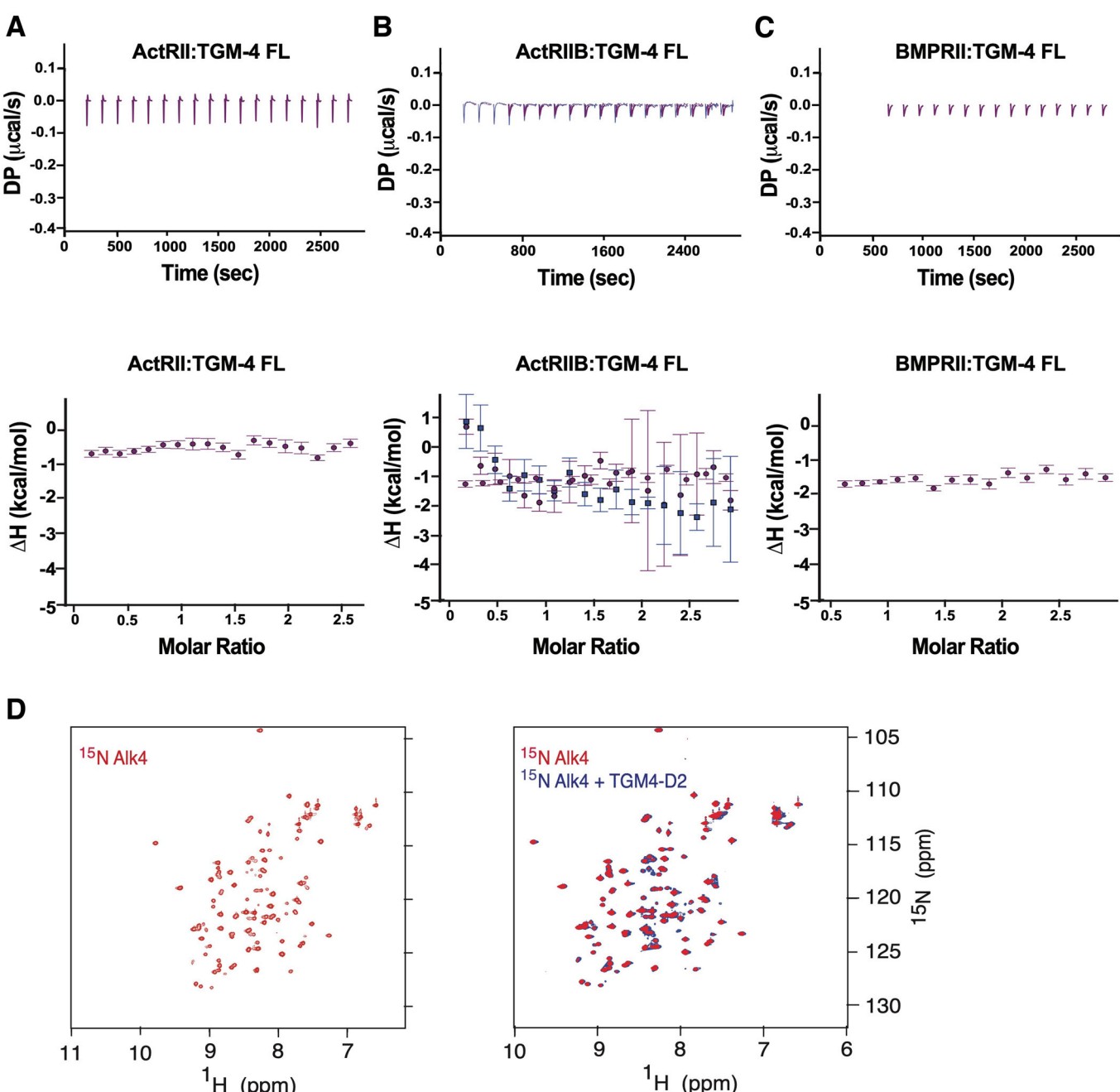

**Figure EV3. Testing binding of TGM4 to other TGF-β family receptors.**

(A) ITC analysis of interaction of ActRII with full-length (FL) TGM4; upper panel presents the raw measured heat represented as differential power (DP) for successive 2.5 µL injections of 200 µM ActRII into a cell containing 300 µL of 15 µM FL TGM4; lower panel presents the integrated heats for these data represented as the change in enthalpy (ΔH) as a function of the increasing molar ratio of ActRII to FL TGM4. One independent measurement, depicted in purple was performed. No binding isotherm could be fit. (B) As (A), for interactions of FL TGM4 with His-tagged ActRIIB; 300 µL of 10 µM FL TGM4 was titrated with successive 2.5 µL injections of 150 µM his-tagged ActRIIB). Two independent measurements, depicted in blue and purple, were performed. No binding isotherm could be fit. (C) As (A), for interactions of FL TGM4 with BMPRII; (300 µL of 10 µM FL TGM4 was titrated with successive 2.5 µL injections of 150 µM BMPRII). One independent measurement, depicted in purple was performed. No binding isotherm could be fit. (D) NMR analysis of TGM4 D2 interaction with ALK4 Type I receptor. ¹H-¹⁵N spectrum of ¹⁵N Alk4 alone (left, red) and overlaid onto the ¹H-¹⁵N spectrum of ¹⁵N Alk4 bound to 1.2 molar equivalents of unlabelled TGM4 D2 (right, blue).

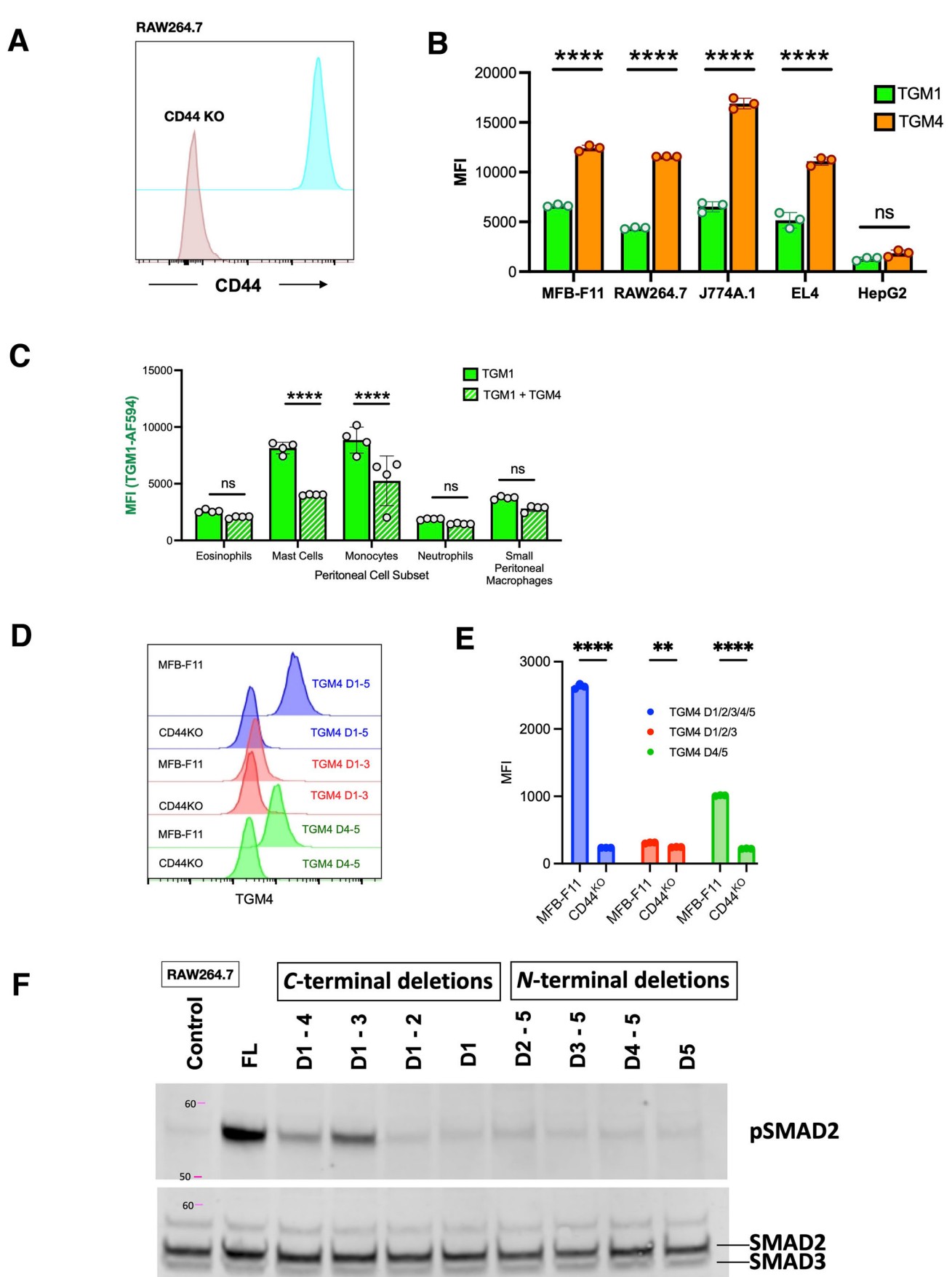

**Figure EV4.  Domain and co-receptor interactions of TGM4.**

(A) Flow cytometric analysis of CD44 binding to control RAW264.7 cells (cyan) and *Cd44*-deleted RAW264.7 cells (tan). (B) Binding of AF594-labelled TGM1 and TGM4 constructs to the indicated cell lines, measured by Mean Fluorescence Intensity on a flow cytometer ($n = 3$). One of 2 biological replicate experiments, $n = 3$, presented as mean ± SD. (C) Quantification of staining of AF594-labelled TGM1 to peritoneal eosinophils (SiglecF$^+$, CD11b$^{(int)}$, Ly6G$^{(low)}$, CD117$^-$), mast cells (CD117$^+$, SSC$^{(high)}$), monocytes (CD11b$^+$, CD115$^+$, CD117$^-$, Ly6G$^-$, F4/80$^-$, SiglecF$^-$, MHC-II$^-$, CD3$^-$, CD19$^-$), neutrophils (CD11b$^+$, Ly6G$^+$, CD117$^-$) and small peritoneal macrophages (CD11b$^+$, CD117$^-$, Ly6G$^-$, F4/80$^{(low)}$, SiglecF$^-$, MHC-II$^{(high)}$, CD3$^-$, CD19$^-$), in absence (solid bars) or presence (hatched bars) of TGM4, as measured by MFI. One of 2 biological replicate experiments, $n = 4$, presented as mean ± SD. (D, E) Flow cytometric analysis of TGM4 binding to MFB-F11 wild-type and CD44-deficient cells, probed with full-length TGM4 D1-5, and truncated constructs D1-3 and D4–5. Example histograms (D) and results from 3 biological replicate experiments (E) are shown. Data shown ($n = 3$), presented as mean ± SD. (F) pSMAD induction in RAW264.7 cells by truncated 10 ng/mLTGM4 constructs, with C-terminal deletions and N-terminal deletions, assessed by western blotting. FL Full-length. Data Information: Data in (B, C, E), analysed by 2-way ANOVA; in (B), each comparison between TGM1 and TGM4, $P < 0.0001$; in (C), comparison between TGM1 and TGM4 for mast cells and macrophages; in (E), comparisons between wild-type and CD44 KO MFB-F11 fibroblasts with TGM4 D1-5 and TGM4 D4–5, both $P < 0.0001$; comparison with TGM4 D1-3, $P = 0.0084$.

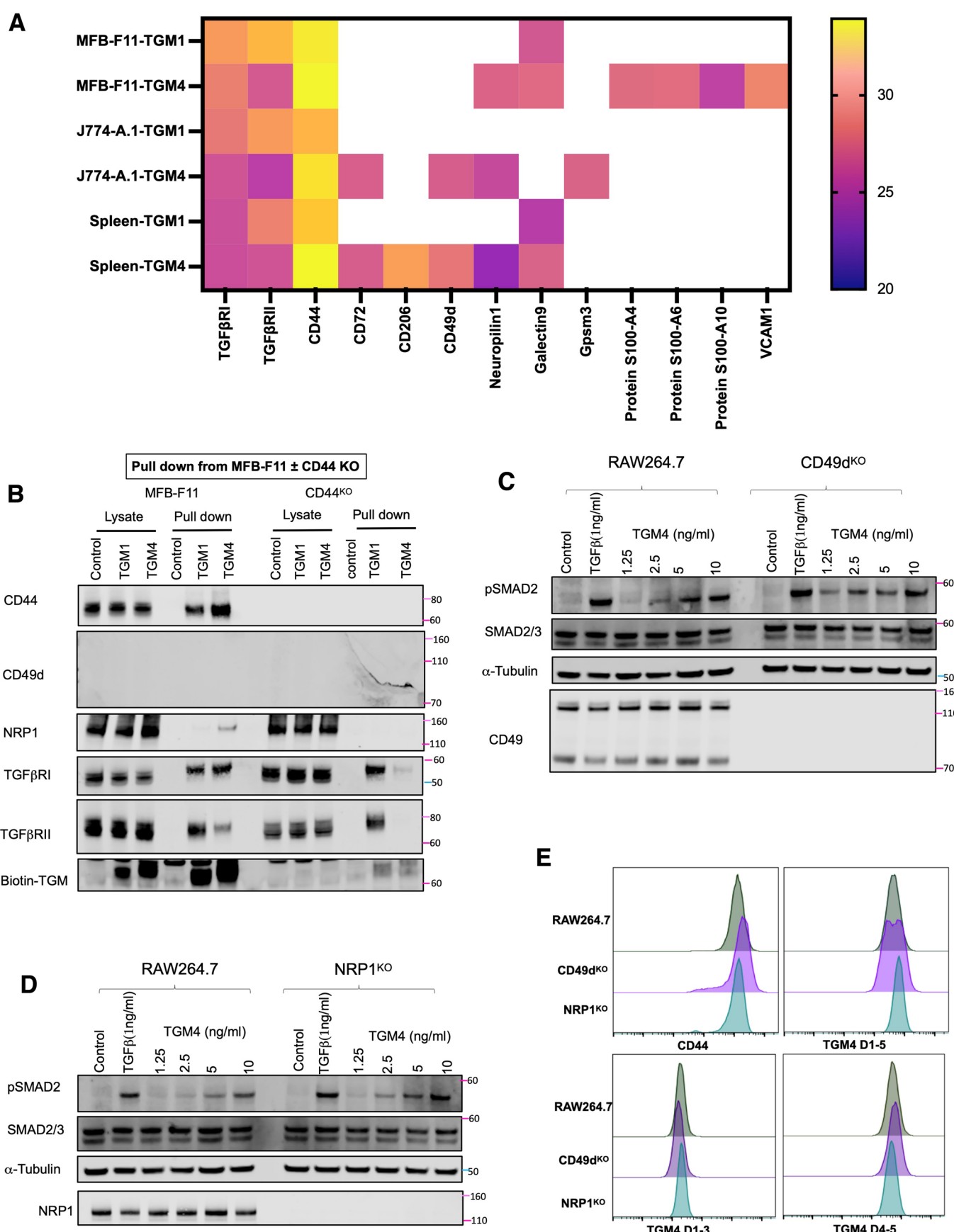

**Figure EV5.  Functional analysis of co-receptors.**

(A) Heat map of mean mass spectrometry Label-Free Quantitation intensity values for proteins detected in each case, in the samples presented in Fig. 5A–F. (B) Pull-down and Western blot analysis in CD44-sufficient and -deficient MFB-F11 cells, probed with antibodies to the indicated proteins. Image is from one of 3 biological replicate experiments. (C) SMAD phosphorylation in RAW264.7 cells sufficient or deficient for CD49d, following stimulation with the indicated concentrations of TGM4. Image is from one of 3 biological replicate experiments. (D) As (B), but with RAW264.7 cells sufficient or deficient for NRP-1. Image is from one of 3 biological replicate experiments. (E) Flow cytometric measurement of TGM4 binding to cells lacking CD49d or NRP-1. RAW264.7 control cells, and sublines in which expression of CD49d or NRP-1 was genetically deleted, were probed by flow cytometry for binding to anti-CD44, TGM4 D-15, D1-3 or D4–5 as indicated.

