## [Peer Review File · EMBO Reports]

The TGF- β mimic TGM4 achieves cell specificity through combinatorial surface co-receptor binding.

Shashi Singh, Danielle Smyth, Kyle Cunningham, Ananya Mukundan, Chang-Hyeock Byeon, Cynthia Hinck, Madeleine White, Claire Ciancia, Natalia Wařowska, Anna Sanders, Regina Jin, Ruby White, Sergio Lilla, Sara Zanivan, Christina Schoenherr, Gareth Inman, Maarten van Dinther, Peter ten Dijke, Andrew Hinck, and Rick Maizels

Corresponding author(s): Rick Maizels (Rick.Maizels@glasgow.ac.uk)

Review Timeline:

Submission Date:	13th Apr 24
Editorial Decision:	15th Apr 24
Revision Received:	16th Aug 24
Editorial Decision:	17th Sep 24
Revision Received:	30th Oct 24
Accepted:	7th Nov 24

Editor: Achim Breiling

Transaction Report: This manuscript was transferred to EMBO reports following peer review at The EMBO Journal.

Dear Dr. Maizels,

Thank you for transferring your manuscript to EMBO reports. I now went through the manuscript and the referee reports from The EMBO Journal (attached again below). The referees have several concerns and suggestions to improve the manuscript, or to strengthen the data and the conclusions drawn.

Given the constructive referee comments, I would like to invite you to revise your manuscript with the understanding that all concerns of the referees must be addressed in the revised manuscript or in a detailed point-by-point response. Acceptance of your manuscript will depend on a positive outcome of another round of review at EMBO reports, using the same referees.

EMBO reports emphasizes novel functional over detailed mechanistic insight. Thus, we will not require addressing points regarding more mechanism experimentally. However, it will be necessary that during final revision you address all points questioning the main conclusions of the study, and all technical concerns, or points regarding the experimental designs, model systems used, or data presentation. Moreover, premature data should be removed, and the presentation improved and streamlined (as outlined by referee #2 - see also his/her points 9 and 10). We would not necessarily require more in vivo evidence (as requested by referee #3), but in case you have such data, we would welcome their addition

1) a .docx formatted version of the final manuscript text (including legends for main figures, EV figures and tables), but without the figures included. Please make sure that changes are highlighted to be clearly visible. Figure legends should be compiled at the end of the manuscript text.

2) individual production quality figure files as .eps, .tif, .jpg (one file per figure), of main figures and EV figures. Please upload these as separate, individual files upon re-submission. Please make sure that all figure panels are called out separately and sequentially in the manuscript text

For more details please refer to our guide to authors:

See also our guide for figure preparation:

Moreover, please consult our guidelines for figure legend preparation:

4) a complete author checklist, which you can download from our author guidelines

(<https://www.embopress.org/page/journal/14693178/authorguide>). Please insert page numbers in the checklist to indicate where the requested information can be found in the manuscript. The completed author checklist will also be part of the RPF.

Please also follow our guidelines for the use of living organisms, and the respective reporting guidelines:
<http://www.embopress.org/page/journal/14693178/authorguide#livingorganisms>

5) that primary datasets produced in this study (e.g. RNA-seq, ChIP-seq and array data) are deposited in an appropriate public database. This is now mandatory (like the COI statement). If no primary datasets have been deposited in any database, please state this in this section (e.g. 'No primary datasets have been generated and deposited').

The accession numbers and database should be listed in a formal "Data Availability" section (placed after Materials & Methods) that follows the model below. Please note that the Data Availability Section is restricted to new primary data that are part of this study.

Data availability

8) Regarding data quantification and statistics, please make sure that the number "n" for how many independent experiments were performed, their nature (biological versus technical replicates), the bars and error bars (e.g. SEM, SD) and the test used to calculate p-values is indicated in the respective figure legends (also for potential EV figures and all those in the final Appendix). Please also check that all the p-values are explained in the legend, and that these fit to those shown in the figure. Please provide statistical testing where applicable. Please avoid the phrase 'independent experiment', but clearly state if these were biological or technical replicates. Please also indicate (e.g. with n.s.) if testing was performed, but the differences are not significant. In case n=2, please show the data as separate datapoints without error bars and statistics.

See also:

<http://www.embopress.org/page/journal/14693178/authorguide#statisticalanalysis>

9) Please note our reference format:

10) We updated our journal's competing interests policy in January 2022 and request authors to consider both actual and perceived competing interests. Please review the policy <https://www.embopress.org/competing-interests> and add a statement declaring your competing interests. Please name that section 'Disclosure and Competing Interests Statement' and add it after the author contributions section.

11) Please add up to five keywords to the manuscript and order the sections like this using these names:

Title page - Abstract - Keywords - Introduction - Results - Discussion - Methods - Data availability section (DAS) - Acknowledgements - Disclosure and Competing Interests Statement - References - Figure legends - Expanded View Figure legends

12) Please make sure that all the funding information is also entered into the online submission system and is complete and similar to the one in the manuscript text file (in the Acknowledgements).

13) We now use CRediT to specify the contributions of each author in the journal submission system. CRediT replaces the author contribution section. Please use the free text box to provide more detailed descriptions. Thus, please do not provide your final manuscript text file with an author contributions section. See also guide to authors:
<https://www.embopress.org/page/journal/14693178/authorguide#authorshipguidelines>

14) We would encourage you to use 'Structured Methods', our new Materials and Methods format. According to this format, the Materials and Methods section should include a Reagents and Tools Table (listing key reagents, experimental models, software and relevant equipment and including their sources and relevant identifiers) followed by a Methods and Protocols section in which we encourage the authors to describe their methods using a step-by-step protocol format with bullet points, to facilitate the adoption of the methodologies across labs. More information on how to adhere to this format as well as downloadable templates (.doc or .xls) for the Reagents and Tools Table can be found in our author guidelines (section 'Structured Methods'):

I look forward to seeing a revised version of your manuscript when it is ready. Please let me know if you have questions or comments regarding the revision.

Kind regards,

Achim

Referee #1:

The paper from the Maizels group is focused on TGF- β mimics that are secreted by the helminth parasite *Heligmosomoides polygyrus*. The work focuses on a comparison of TGM1 and TGM4. The authors show that whereas TGM1 can activate the TGF- β pathway in a variety of cell types due to its affinity for the TGF- β type I and type II receptors, TGM4 has a more cell type restrictive profile and predominantly activates TGF- β signaling in myeloid cells. The authors show that this is because it has a much weaker affinity for the TGF- β type II receptor and in addition, binds to CD44 and to CD49d (integrin $\alpha 4$) and CD206 at the cell surface. As a result, the authors show that TGM4 can modulate macrophage responses to IL-4 and lipopolysaccharide (LPS).

In general, the work is very well done, and the paper is well written. The biochemistry is very strong, and the data are clear on the differences in binding between TGM1 and TGM4 that explain the different cell type specificity of these two TGF- β mimics. However, I think that the paper does not make a sufficient advance over the same group's PNAS paper last year (PMID 37590410). In that paper the authors showed that CD44 acts as a coreceptor for TGM1 to enhance cell-specific signaling. This current paper shows that this is also true for TGM4, which also binds additional receptors. However, they have not shown that these additional receptors are required for the cell responses. Moreover, I think that more functional data is required in the current paper to reveal the functional consequences of the different cell type specificity between TGM1 and TGM4. Furthermore, what relevance does this have for parasite infection? This needs to be addressed.

I have other specific points below.

1. The authors refer to the IPs in Fig 2c, d quantitatively, but the data have not been quantitated. They should either quantitate several replicates or just rely on the SPR data for a quantitative assessment.
2. The authors show that TGM4 binds to TGFBR2 much more weakly than does TGM1, but provide no mechanistic insights as to why this is the case. This needs to be addressed as it is a key difference between the two TGMs.
3. The authors show that in CD4+ T cells, TGM4 needs an extended duration to induce signaling. The underlying mechanism needs to be determined for this.
4. From the data in Figure 3b the authors conclude there is more CD44 in RAW264.7 cells compared with MFB-F11 cells (line 176/177). I do not think that they can draw this conclusion.
5. The authors show that TGM4 binds additional receptors compared with TGM1, but the functional relevance of this is not demonstrated. This needs to be addressed.
6. In the model in Figure 8, the authors draw CD49d as contacting domain 1 and CD206 as contacting domain 5, but the authors

do not provide any evidence for this. They need to map these interactions in more detail.

Minor points

1. The authors should use the HGNC approved names for all the proteins and genes mentioned. For example, the TGF- β type I and type II receptors should be TGFBR1 and TGFBR2 etc.
2. The authors mention a CD44 knockout line on page 7 line 196 and then explain that they made it in lines 222/223. Is it the same line? If so they should refer to supp Fig 4C when they first mention it.
3. The ITC data in Supp Figure 3a/b needs explaining in more detail.

Referee #2:

This manuscript extends a series of remarkable findings by Dr. Maizels and colleagues on the TGM family, a family of structurally unrelated TGF- β mimics produced by a murine intestinal helminth ostensibly to suppress host immune and inflammatory responses to benefit the parasite. Prior reports by this group elucidated the basis for the interaction of TGM1 with the two TGF- β receptor components (TbR1 and TbR2) and the activation of TGF- β signaling in immune and other cell types. TGM proteins have 5 modular domains and the authors recently reported that domains 1-3 bind to TbR1 and TbR1 while domains 4-5 bind to the hyaluronan receptor CD44, which potentiates the activation of TGF- β signaling. In this new report, the authors show that TGM4 is even more dependent on CD44 for activation of TGF- β signaling. Compared to TGM1, TGM4 has higher affinity for CD44, which compensates for a poor affinity of TGM4 for TbR1, resulting in a TGM that selectively acts on CD44-rich macrophages to suppress inflammation.

Although this work adds a relatively simple twist to the story, this is a remarkable saga of convergent evolution of a parasite genome leveraging the immune suppressive and anti-inflammatory action of the TGF- β pathway in its host. Therefore, the work is of potential interest if properly revised.

On the negative side, this manuscript suffers from a poor and confusing presentation of a heterogeneous mix of assay systems, a switching between different methods, reagents, and illustrations. It also suffers from the inclusion of negative results on membrane proteins that TGM1 and TGM4 bind but play no known role in TGM biology. This gives the impression that several collaborating labs performed partially related experiments using different methods at different times, and then combined a poorly curated dataset to generate their TGM4 manuscript. This manuscript needs a major round of polishing, in addition to addressing various experimental shortcomings to meet normal standards for publication.

Experimental deficiencies:

1. Fig. 1d: The authors used a multi-receptor kinase inhibitor to show that TbR1 mediates TGM4 signaling. Instead of this pharmacological inhibitor, CRISPR-Cas9 KO of TbR1 should be used to make this crucial point. TbR2 KO should also be included.
2. Fig 3a: similar TbR1 signals were obtained in the TGM1 and TGM4 pull downs in MFB-F11 fibroblasts. This doesn't seem to be consistent with the author's claim that TGM4 is a weak TGF- β agonist in these cells compared to TGM1?
3. Supp Fig. 3a,b: This analysis should also include ActRIIB for completion.
4. Does TGM4 form a TbR1-TbR2 dimer as shown in Fig 8 or a (TbR1-TbR2)₂ tetramer like TGF- β does? This could be addressed by transducing cells with different tagged TbR1 constructs engineered with different tags A and B, and then testing the ability of TGM4 addition to form a complex containing tags A and B. Ditto for TbR2.
5. CD44 is necessary for TGM4 activation of TGF- β signaling, but is it sufficient? Do fibroblasts become responsive to TGM4 when transduced with a CD44 vector? Does TGM4 activate TGF- β signaling in any other cell type expressing CD44?

Presentation deficiencies

6. Throughout the text and figures, specify what specific cell types or cell lines were used in each experiment. For example, line 143 should say MFB-F11 fibroblasts and not just "cells". In Figure 4 a, c and e, should say what macrophages were used (i.e. peritoneal, or tissue resident and in what tissue, peritoneal, RAW264.7 cell line).
7. Throughout the text, explain why different types of macrophages were used in different experiments.
8. Throughout the text and figures, explain better why different assays were used in different experiments investigating the TGM-receptor binding interactions.

Premature/irrelevant parts:

9. Fig 5 a-c: these pull-down and mass spec data show that recombinant TGMs interact with an assortment of membrane proteins, but most of these interactions are relatively weak and are no match for the TGM interaction with CD44. The authors tried but were unable to show that any of these other interactions matter in TGM biology and TGM activation of TGF- β signaling. For all we know, these could all be background interactions in cells expressing recombinant TGMs (or however these pull-downs were done; the text and figure legend don't say).
10. Given this, the following should be removed: the mention of CD49b/Itga4 and CD206 in the Abstract; the entire Figure 6 and associated text (lines 245-283) or replace it with a brief statement; lines 355-370 in the Discussion; and CD49b and CD206 in Fig 8.

Minor:

11. Fig 1e: why is it that HepG2 cells respond to TGF- β but not to TGM1?
12. Fig 2a: the text should say that the p-Smad2 effect of TGM1 1-3 is also "muted".
13. Line 158: "no more than moderate affinity" is a scientifically vague phrase.
14. Fig 7: explain why different cytokines were studied in different macrophages
15. Fig 3b-d, Y axis: "Relative fold change", What change? And relative to what?
16. Fig 5 a-c: be consistent in the use of alternative names for Mrc1/CD206 and Itga4/CD49b in the figure vs the text.

Referee #3:

In this manuscript, Singh et al. describe an intriguing family of TGF- β mimic proteins (TGMs) produced by the helminth parasite *Heligmosomoides polygyrus* that can selectively modulate host immune cells, particularly myeloid populations like macrophages. The key finding is that one member, TGM4, displays cell-type specificity compared to TGF- β itself by virtue of combinatorial binding to both TGF- β receptors as well as additional co-receptors like CD44, CD49d and CD206 predominantly found on myeloid cells. This multi-receptor engagement allows TGM4 to specifically modulate myeloid cells, without affecting fibroblasts and only minimally impacting T cells.

The principal significance lies in elucidating the molecular mechanisms underlying TGM4's myeloid cell specificity, offering insights into how pathogens manipulate host immune responses. However, more direct *in vivo* evidence of TGM4 selectively modulating myeloid cells is needed. Additionally, the manuscript requires polishing - statements need clarification, figures require clearer presentation, labeling, proper controls, and statistical analyses.

Major Concerns:

1. The key conclusion that TGM4 selectively targets myeloid cells is based primarily on *in vitro* data using cell lines and some primary cells. Stronger *in vivo* evidence validating this specificity would greatly strengthen the findings.
2. While the study highlights TGM4's unique properties compared to TGM1, the functional differences between them in modulating macrophages (Fig. 7) are unclear despite claims of differential co-receptor binding.
3. The functional relevance of each co-receptor in mediating TGM4 effects is not clearly delineated. For example, is the lower TGF β 2 affinity critical for downstream activation? Genetic deletion of some co-receptors did not ablate signaling, requiring better clarification of their precise roles.

Minor Concerns:

1. Line 95 states TGM7 was inactive on MFB-F11 cells, but Fig 1a lacks TGM7 data for this cell line.
2. Internal controls for Western blots are missing, such as in Figure 1g, which undermines the ability to discern if the observed changes in pSMAD2 levels are due to the change of SMAD2 or protein loading.
3. There are inconsistencies between Fig 3a-b (stronger CD44 binding by TGM4 compared to TGM1) and Fig 3f (similar CD44 binding compared to TGM1).
4. Figure labels, particularly in Figures 4e-4f, are unclear, making it difficult to interpret which lines correspond to which concentrations.
5. Many figures lack statistical analyses.

Additional Suggestions:

1. Examining *in vivo* effects of neutralizing/blocking TGM4 during *H. polygyrus* infection would provide insights into its physiological role.
2. Testing TGM4 in pre-clinical disease models could explore its therapeutic potential for selectively targeting myeloid cells.

Re : Singh *et al.*

The TGF- β mimic TGM4 achieves cell specificity through combinatorial surface co-receptor binding.

Authors' Response to Reviewers.

We thank each of the reviewers for their critical yet constructive comments which we believe have resulted in a significantly improved manuscript, and hope that our responses detailed below, will meet with their full approval.

Reviewer 1

The paper from the Maizels group is focused on TGF- β mimics that are secreted by the helminth parasite *Heligmosomoides polygyrus*. The work focuses on a comparison of TGM1 and TGM4. The authors show that whereas TGM1 can activate the TGF- β pathway in a variety of cell types due to its affinity for the TGF- β type I and type II receptors, TGM4 has a more cell type restrictive profile and predominantly activates TGF- β signaling in myeloid cells. The authors show that this is because it has a much weaker affinity for the TGF- β type II receptor and in addition, binds to CD44 and to CD49d (integrin $\alpha 4$) and CD206 at the cell surface. As a result, the authors show that TGM4 can modulate macrophage responses to IL-4 and lipopolysaccharide (LPS).

In general, the work is very well done, and the paper is well written. The biochemistry is very strong, and the data are clear on the differences in binding between TGM1 and TGM4 that explain the different cell type specificity of these two TGF- β mimics. However, I think that the paper does not make a sufficient advance over the same group's PNAS paper last year (PMID 37590410). In that paper the authors showed that CD44 acts as a coreceptor for TGM1 to enhance cell-specific signaling. This current paper shows that this is also true for TGM4, which also binds additional receptors. However, they have not shown that these additional receptors are required for the cell responses. Moreover, I think that more functional data is required in the current paper to reveal the functional consequences of the different cell type specificity between TGM1 and TGM4. Furthermore, what relevance does this have for parasite infection? This needs to be addressed.

We thank the Reviewer for these broadly positive remarks, and are pleased to respond to the specific points below. We have understood from the Editor that further functional data are not required at this stage, with a manuscript aiming to establish the concept of combinatorial signalling as a strategy for cell specificity. We are indeed most interested in the functional consequences which are the subject of a new phase of our work. With respect to the relevance for parasite infection, we have studied CD44-deficient mice which show no greater resistance to infection but in a context of a severely perturbed immune system, and plan to test CD49d and CD206-deficient mice. We do not yet have access to conditional (ie myeloid-cell restricted) KO mice, which would be ideal to address the Reviewer's question.

I have other specific points below.

1. The authors refer to the IPs in Fig 2c, d quantitatively, but the data have not been quantitated. They should either quantitate several replicates or just rely on the SPR data for a quantitative assessment.

This experiment presented in this Figure was conducted 3 times, and we have now quantified intensities, confirming statistically significant differential binding, TGM4 more strongly to T β RI and TGM1 more to T β RII, which we present in a revised Figure 2 panels d and f.

2. The authors show that TGM4 binds to TGFBR2 much more weakly than does TGM1, but provide no mechanistic insights as to why this is the case. This needs to be addressed as it is a key difference between the two TGMs.

The Reviewer raises an important point; although we do not yet have a crystal structure for TGM4, comparison with TGM1 shows that the Lys-254 and Asn-255 dipeptide which contains one basic residue likely to form an ion bridge with Asp-55 of T β RII are substituted a Ser-His dipeptide. We now include this information in the text (lines 364-367).

3. The authors show that in CD4+ T cells, TGM4 needs an extended duration to induce signaling. The underlying mechanism needs to be determined for this.

We thank the Reviewer for raising this point, revealed by our Imagestream analysis, and paralleling earlier studies with TGM1, which showed slower but more sustained signaling compared to TGF- β (White et al, Ref.11). We surmise that the slower kinetics may reflect assembly of a multimeric and longer-lasting signalling complex, but without further data would prefer not to speculate in this manuscript.

4. From the data in Figure 3b the authors conclude there is more CD44 in RAW264.7 cells compared with MFB-F11 cells (line 176/177). I do not think that they can draw this conclusion.

The Reviewer is correct to raise this point, which is poorly phrased in the original manuscript: this now reads (lines 187-188) "Quantification from replicate Western blot experiments showed that *TGM4 co-precipitated considerably more CD44 from RAW264.7 cells than did TGM1, and a similar trend was observed with MFB-F11 cells.*"

5. The authors show that TGM4 binds additional receptors compared with TGM1, but the functional relevance of this is not demonstrated. This needs to be addressed.

We accept that functional roles for the additional receptors (CD49d, CD206, Nrp1) have yet to be established. We do show in EV Figure 5b and c (Figure 6 d,e in the original submission) that SMAD2 signalling is intact in cells lacking CD49d or Nrp1 but further investigation is required to ascertain if these co-receptors function to enhance avidity and/or stabilise receptor complexes, or if they mediate a separate signalling pathway in their own right. We are now planning such functional studies but understand from the Editor that we are not expected to include them in the revised version of the manuscript. We hope that this will meet with the Reviewer's agreement.

6. In the model in Figure 8, the authors draw CD49d as contacting domain 1 and CD206 as contacting domain 5, but the authors do not provide any evidence for this. They need to map these interactions in more detail.

We appreciate the Reviewer's comment and accept that the model overstates the evidence at hand, which is based on Figure 5h (Figure 6c in the original submission) showing that D1/2/3 bind CD49d, and not D1 alone, and D4/5 binding CD206, and not D5 alone. As the precise domain interactions with each co-receptor have yet to be determined, we have decided to omit this Figure from the revised submission.

Minor points

1. The authors should use the HGNC approved names for all the proteins and genes mentioned. For example, the TGF- β type I and type II receptors should be TGFBR1 and TGFBR2 etc.

With respect, the nomenclature we use is more generic for mouse (which is the source of all cells shown except HepG2) and human, and widely used in numerous contemporary publications.

2. The authors mention a CD44 knockout line on page 7 line 196 and then explain that they made it in lines 222/223. Is it the same line? If so they should refer to supp Fig 4C when they first mention it.

We thank the Reviewer for catching this inadvertent duplication. We have removed the second mention (lines 240-241 in the revised manuscript) and cite the Figure (now Figure EV 4 a) at the first instance (line 210).

3. The ITC data in Supp Figure 3a/b needs explaining in more detail.

We have now expanded the Legend for Figure EV3 to define the abbreviations (DP, differential power; ΔH , change in enthalpy) and to specify volumes and concentrations of ligands titrated into the ITC cells.

Reviewer 2

This manuscript extends a series of remarkable findings by Dr. Maizels and colleagues on the TGM family, a family of structurally unrelated TGF- β mimics produced by a murine intestinal helminth ostensibly to suppress host immune and inflammatory responses to benefit the parasite. Prior reports by this group elucidated the basis for the interaction of TGM1 with the two TGF- β receptor components (TbR1 and TbR2) and the activation of TGF- β signaling in immune and other cell types. TGM proteins have 5 modular domains and the authors recently reported that domains 1-3 bind to TbR1 and TbR1 while domains 4-5 bind to the hyaluronan receptor CD44, which potentiates the activation of TGF- β signaling. In this new report, the authors show that TGM4 is even more dependent on CD44 for activation of TGF- β signaling. Compared to TGM1, TGM4 has higher affinity for CD44, which compensates for a poor affinity of TGM4 for TbR1, resulting in a TGM that selectively acts on CD44-rich macrophages to suppress inflammation.

Although this work adds a relatively simple twist to the story, this is a remarkable saga of convergent evolution of a parasite genome leveraging the immune suppressive

and anti-inflammatory action of the TGF- β pathway in its host. Therefore, the work is of potential interest if properly revised.

On the negative side, this manuscript suffers from a poor and confusing presentation of a heterogeneous mix of assay systems, a switching between different methods, reagents, and illustrations. It also suffers from the inclusion of negative results on membrane proteins that TGM1 and TGM4 bind but play no known role in TGM biology. This gives the impression that several collaborating labs performed partially related experiments using different methods at different times, and then combined a poorly curated dataset to generate their TGM4 manuscript. This manuscript needs a major round of polishing, in addition to addressing various experimental shortcomings to meet normal standards for publication.

We thank the Reviewer for their considered, and in parts complimentary, summary. We apologise if the presentation comes across in any way confusing, as our intention in testing multiple cell types in different systems in the different laboratories was to establish a robust case that was not dependent on the vagaries of any individual cell line. We feel the Reviewer's wording is a little harsh in terms of our collaboration and the integration of results, but nevertheless have endeavoured to follow the recommendations to clarify both the components and the rationale for each experimental setting as below.

Experimental deficiencies:

1. Fig. 1d: The authors used a multi-receptor kinase inhibitor to show that TbR1 mediates TGM4 signaling. Instead of this pharmacological inhibitor, CRISPR-Cas9 KO of TbR1 should be used to make this crucial point. TbR2 KO should also be included.

We appreciate the Reviewer's point here, and as stated in the text (lines 108-109), we recognise that SB431542 can also interfere with the ALK4 and ALK7 kinases, as well as T β RI (ie ALK5). However, we feel that activation through T β RI is addressed with unequivocal biophysical data later in the paper showing high affinity TGM4 interaction with ALK5 and not with ALK4, which is more closely related to ALK5 than is ALK7. As our data do not explicitly exclude interactions with the more distant ALK7, which responds to activin rather than TGF- β , we have amended the wording (line 107) to refer to the "SMAD" signalling pathway, in place of the "TGF- β " pathway, and moved the graph to the supplementary information in Figure EV 1 c.

2. Fig 3a: similar TbR1 signals were obtained in the TGM1 and TGM4 pull downs in MFB-F11 fibroblasts. This doesn't seem to be consistent with the author's claim that TGM4 is a weak TGF- β agonist in these cells compared to TGM1?

The Reviewer raises a point we consider to be central to the story; binding to T β RI is in itself insufficient to activate signalling in fibroblasts, even if pull-downs show an apparent strong interaction. A similar examples can be seen with HepG2 cells in which a strong T β RI band is seen in pull-down (right handpanel in Fig 3 a) but no SMAD activation (as shown in Fig 1 d, far right panel).

3. Supp Fig. 3a,b: This analysis should also include ActRIIB for completion.

We thank the Reviewer for this suggestion, which we have now tested, with no binding of TGM4 found to ActRIIB; we have added these data to Figure EV 3 b, with appropriate additions to the text (lines 171-172).

4. Does TGM4 form a TbR1-TbR2 dimer as shown in Fig 8 or a (TbR1-TbR2)₂ tetramer like TGF- β does? This could be addressed by transducing cells with different tagged TbR1 constructs engineered with different tags A and B, and then testing the ability of TGM4 addition to form a complex containing tags A and B. Ditto for TbR2.

This is a very interesting question. By size exclusion chromatography (SEC) the protein behaves as a monomer and there are no available cysteines to form a TGF- β -like dimer. But formation of a non-covalent dimer under physiological conditions, especially after receptor binding, remains a possibility. We plan to first address this through cryo-EM of the TGM4-T β RI-T β RII complex, and gain insight into whether higher order multimers are observed.

5. CD44 is necessary for TGM4 activation of TGF- β signaling, but is it sufficient? Do fibroblasts become responsive to TGM4 when transduced with a CD44 vector? Does TGM4 activate TGF- β signaling in any other cell type expressing CD44?

In fact, MFB-F11 fibroblasts do express a low level of CD44, so that we believe there is a threshold which must be crossed for activation to take place; as we stated in the original text (lines 372-375) "*It is likely that the selectivity of TGM4 is based on a continuum threshold; although MFB-F11 fibroblasts do express the CD44 co-receptor (and can bind TGM4 in flow cytometry assays), either the expression level is too low, and/or the additional coreceptors required for high affinity interactions are absent, resulting in the failure of TGM4 to assemble an activation complex*". It would be interesting to test this with a titratable induction system and we are exploring this possibility.

Presentation deficiencies

6. Throughout the text and figures, specify what specific cell types or cell lines were used in each experiment. For example, line 143 should say MFB-F11 fibroblasts and not just "cells". In Figure 4 a, c and e, should say what macrophages were used (i.e. peritoneal, or tissue resident and in what tissue, peritoneal, RAW264.7 cell line).

We thank the Reviewer for these suggestions which we have adopted throughout the manuscript, replacing "cells" with more specific designations, which certainly improve the clarity of the presentation.

7. Throughout the text, explain why different types of macrophages were used in different experiments.

Line 120 : Added "In addition to cell lines" (*.we investigated responses of mouse primary cells...*)

Lines 207-208, "*we chose to study RAW264.7 macrophages as these can be modified by CRISPR-Cas9 gene editing*"

Lines 224-225 : "*To test the relative strength of TGM1 and TGM4 interactions across a range of physiological cell types, we evaluated binding to primary peritoneal exudate cells*"

8. Throughout the text and figures, explain better why different assays were used in different experiments investigating the TGM-receptor binding interactions.

As above on Page 8 we now state “*To test the relative strength of TGM1 and TGM4 interactions across a range of physiological cell types, we evaluated binding to primary peritoneal exudate cells...*”

As discussed below, we also have added explanations of the pulldown methodology (as above) and the knockout cell types.

Premature/irrelevant parts:

9. Fig 5 a-c: these pull-down and mass spec data show that recombinant TGMs interact with an assortment of membrane proteins, but most of these interactions are relatively weak and are no match for the TGM interaction with CD44. The authors tried but were unable to show that any of these other interactions matter in TGM biology and TGM activation of TGF- β signaling. For all we know, these could all be background interactions in cells expressing recombinant TGMs (or however these pull-downs were done; the text and figure legend don't say).

We first apologise for not clearly setting out the pull-down methodology used for mass spectrometry presented in Figure 5. Here, purified biotinylated proteins were incubated with target cells, and pull-downs used streptavidin beads followed by stringent washes to remove nonspecific contaminants; as we included primary splenocytes as a target population, endogenous transfection would not have been a realistic option in any event. It is interesting to note that with the same cells under identical conditions, TGM4 pulls down additional proteins, and key additional proteins (Itga4, =C49d), which are found on both J774.1 cell line macrophages and primary splenocytes; moreover, the intensity of the newly identified co-receptors (including Mrc1=CD206 and Nrp1) is comparable to T β RI and T β RII, supporting the case that these are genuine interactors, as subsequently validated by Western blotting (now in 5 g, h; previously in Figure 6). The superlative signal for CD44 reflects, we suggest, its abundance relative to the other receptors in question.

We have added (lines 267-268) the clarification *adding exogenous biotin-labelled TGM proteins to cell suspensions and performing pull-downs with streptavidin beads,* and lines 285-286 “*incubated with exogenous biotinylated TGM proteins followed by streptavidin affinity purification.*” We have also added to the Figure 5 Legend “*Pull-down samples purified from cells treated with biotin-tagged TGM4 and precipitated with streptavidin resin....*”

10. Given this, the following should be removed: the mention of CD49b/Itga4 and CD206 in the Abstract; the entire Figure 6 and associated text (lines 245-283) or replace it with a brief statement; lines 355-370 in the text; and CD49b and CD206 in Fig 8.

We understand the Reviewer's suggestion, but feel it arose from their concern that the evidence for CD49b/Itga4 and CD206 was based on endogenous expression of TGM transgenes within the cells in question. In fact, as explained above, each of the datasets were derived from exogenously applied biotin-tagged TGM proteins, and precipitated with streptavidin resin. The mass spec experiments were performed on primary splenocytes as well as two cell lines (fibroblasts and macrophages) and

compared TGM1 and TGM4. From this analysis, CD206 (Mrc1) was second only to CD44 in spleen cells with TGM4; and CD49b/Itga4 significant in both spleen cells and J774.1 macrophages; hence both are strongly supported by mass spec data.

Importantly we validated these candidates by Western blotting in data presented in the original Figure 6; hence we feel it is appropriate to mention these co-receptors in the Abstract and in the text. As mentioned above, we now omit the Figure 8 schematic from the submission.

Minor:

11. Fig 1e: why is it that HepG2 cells respond to TGF- β but not to TGM1?

HepG2 cells do not express CD44 (Figure 3a, far right panel), which may account for the inability of TGMs to activate signalling in these cells, at least at the doses used in our experiments.

12. Fig 2a: the text should say that the p-Smad2 effect of TGM1 1-3 is also "muted".

We have rephrased this text (lines 148-149) to be clearer, now stating "Notably, for both TGM1 and TGM4 activity was attenuated in forms lacking D4-5".

13. Line 158: "no more than moderate affinity" is a scientifically vague phrase.

We have reworded this (line 168) to simply state "relatively low binding affinity" as shown by SPR, as a prelude to the investigation of alternative receptors for TGM4.

14. Fig 7: explain why different cytokines were studied in different macrophages

Different macrophage populations characteristically respond to different ligands (LPS, IL-4) by expression of different markers and cytokines; in particular cell lines show a restricted profile, and bone marrow-derived macrophages a broader set of responses.

15. Fig 3b-d, Y axis: "Relative fold change", What change? And relative to what?

In these figures, the lowest groups have been normalised to 1, and the fold changes are relative to that. To clarify we have added this wording to the Figure Legend : Data are shown as fold change relative to TGM1 (b, c) or TGM4 (d) that have been respectively normalised to 1.

16. Fig 5 a-c: be consistent in the use of alternative names for Mrc1/CD206 and Itga4/CD49b in the figure vs the text.

We have amended the annotations in Figure 5 a and c, and in the text to use CD49d and CD206 as the primary designations. The gene names Mrc1 and Itga4 are now only used in the text at first introduction (line 270).

Reviewer 3:

In this manuscript, Singh et al. describe an intriguing family of TGF- β mimic proteins (TGMs) produced by the helminth parasite *Heligmosomoides polygyrus* that can

selectively modulate host immune cells, particularly myeloid populations like macrophages. The key finding is that one member, TGM4, displays cell-type specificity compared to TGF- β itself by virtue of combinatorial binding to both TGF- β receptors as well as additional co-receptors like CD44, CD49d and CD206 predominantly found on myeloid cells. This multi-receptor engagement allows TGM4 to specifically modulate myeloid cells, without affecting fibroblasts and only minimally impacting T cells.

The principal significance lies in elucidating the molecular mechanisms underlying TGM4's myeloid cell specificity, offering insights into how pathogens manipulate host immune responses. However, more direct in vivo evidence of TGM4 selectively modulating myeloid cells is needed. Additionally, the manuscript requires polishing - statements need clarification, figures require clearer presentation, labeling, proper controls, and statistical analyses.

We appreciate the Reviewers' positive comments and also thank them for presentational suggestions which we have followed as detailed below.

Major Concerns:

1. The key conclusion that TGM4 selectively targets myeloid cells is based primarily on in vitro data using cell lines and some primary cells. Stronger in vivo evidence validating this specificity would greatly strengthen the findings.

We very much agree with the need for in vivo studies to elaborate and extend on our findings; these are under way but require careful and detailed investigations, for example to cover a comprehensive range of doses and routes of administration, with multiple time points and, most importantly, appropriate readouts. For example, preliminary experiments have tested TGM4's effect on resident peritoneal macrophages (as shown in Figure 6 in the revised submission), using a single dose and time point; the effects on resting myeloid cells may not be as important as effects on inflammatory cells for which the range of options is quite extensive. Hence we feel that this inquiry will represent a full manuscript in its own right.

2. While the study highlights TGM4's unique properties compared to TGM1, the functional differences between them in modulating macrophages (Fig. 7) are unclear despite claims of differential co-receptor binding.

We agree with the reviewer that both TGM ligands can modulate macrophages, and that the data in Figure 7 show only minor differences in effect on this cell type; in experiments recently initiated we have uncovered differential signalling of non-SMAD pathways (through S6 phosphorylation), and a greater inhibition of antigen processing by TGM4; however, these studies are ongoing and extend beyond the scope of the current manuscript. We allude to the possibility of additional signalling in the original text (lines 379-381).

3. The functional relevance of each co-receptor in mediating TGM4 effects is not clearly delineated. For example, is the lower TGF β R2 affinity critical for downstream activation? Genetic deletion of some co-receptors did not ablate signaling, requiring better clarification of their precise roles.

The Reviewer raises an important issue (as also raised by Reviewer 1 in point #5) which reflects the calibrated nature of these receptor interactions, governed by thresholds rather than a binary presence or absence of a particular co-receptor. In the case of CD44, we found the absence of this receptor reduced potency by an order of magnitude, but did not abolish signalling altogether. Hence, the role of each co-receptor needs to be investigated in the context of a range of ligand doses with cell types expressing differing receptor densities. In response to the question of the lower affinity for T β R2, we clearly show that this compromises the ability of TGM4 to activate fibroblasts (which have low CD44 levels and lack CD49d or CD206), but does not ablate activation of macrophages which express all these markers at significant levels.

Minor Concerns:

1. Line 95 states TGM7 was inactive on MFB-F11 cells, but Fig 1a lacks TGM7 data for this cell line.

We have published the lack of activity of TGM7 (Ref. 15, Smyth 2017, Figure 3) albeit at a single concentration; in Fig 1a we extend the dose range of TGM4 to allow comparison with its activity on T cells. We now cite Ref 15 to indicate that this information has been published.

2. Internal controls for Western blots are missing, such as in Figure 1g, which undermines the ability to discern if the observed changes in pSMAD2 levels are due to the change of SMAD2 or protein loading.

We included tubulin controls in earlier experiments, although where the interval between ligand addition and cell lysis for Western blotting was no more than 60 minutes, experience indicates total SMAD2 levels do not change within that time frame. To respond to the Reviewer's concern, we now show the tubulin controls for Figure 1 d (in which 5 different cell lines were tested), that supports the conclusion that total SMAD2 levels are essentially constant within the time limits of this protocol.

3. There are inconsistencies between Fig 3a-b (stronger CD44 binding by TGM4 compared to TGM1) and Fig 3f (similar CD44 binding compared to TGM1).

The Reviewer raises an important point that we have also noted, as in Figure 3a-b full-length TGMs are used in cell assays, while the biophysical determinations in Fig 3f have tested Domains 4 -5 alone. One interpretation is that Domains 1-3 of TGM4 contribute to overall CD44 binding affinity, either through direct binding or some form of co-operativity; we will address this question once the structure of TGM4:CD44 complexes are resolved.

4. Figure labels, particularly in Figures 4e-4f, are unclear, making it difficult to interpret which lines correspond to which concentrations.

In these panels, a single concentration of fluorescent ligand is used and the graphs represent fluorescent intensity on individual cells. We have reorganised the Figure to separate out the composite graphs and make the labelling clearer, with corresponding changes to the Figure Legend, which we hope will be to the Reviewer's satisfaction.

5. Many figures lack statistical analyses.

We apologise for omitting some of the statistical analyses, these are now provided in Figure 1 e (4 panels), 3 I and throughout Figure 6 (the previous Figure 7). In the very first panels (Figure 1 a-c) where n is low and effect sizes are vary large, we have not applied statistical analyses.

Additional Suggestions:

1. Examining in vivo effects of neutralizing/blocking TGM4 during H. polygyrus infection would provide insights into its physiological role.

Currently, we only have monoclonals to TGM1, with some cross-reactivity to TGM4, but if we can produce a TGM4-specific antibody, this would be an important experiment.

2. Testing TGM4 in pre-clinical disease models could explore its therapeutic potential for selectively targeting myeloid cells.

We have performed this in airway allergy and colitis but wish to characterise the mechanistic pathway (e.g. tolerant myeloid cells vs regulatory T cells) in an in-depth study beyond the scope of the current manuscript.

In concluding, we would like to again express our thanks to all Reviewers for their insightful and constructive comments.

Dear Dr. Maizels

Thank you for the submission of your revised manuscript to our editorial offices. I have now received the reports from two of the three referees that I asked to re-evaluate the study, you will find below. Original referee #1 was unresponsive to my invitations to re-assess the manuscript. However, going through your point-by-point-response, I consider his/her points as adequately addressed. As you will see, the other two referees now fully support the publication of the study in EMBO reports.

Before we can proceed with formal acceptance, I have these editorial requests I ask you to address in a final revised manuscript:

- Please remove the one sentence summary from the title page. This can be re-used for the synopsis blurb (see below).
- Please reduce the number of keywords to 5 and order the manuscript sections like this, using these names: Abstract - Keywords - Introduction - Results - Discussion - Methods - Data availability section - Acknowledgements - Disclosure and Competing Interests Statement - References - Figure legends - Tables (with legends) - Expanded View Figure legends
- There is a last name discrepancy. It is 'Natalia Wasowska' in the manuscript, but 'Natalia Wasowksa' in the submission system. Please check.
- Please add corr. author name, journal name and the manuscript ID# to the author checklist.
- Please make sure that the number "n" for how many independent experiments were performed, their nature (biological versus technical replicates), the bars and error bars (e.g. SEM, SD) and the test used to calculate p-values is indicated in the respective figure legends. Please also check that all the p-values are explained in the legend, and that these fit to those shown in the figure. Please provide statistical testing where applicable. Please avoid the phrase 'independent experiment', but clearly state if these were biological or technical replicates. Please also indicate (e.g. with n.s.) if testing was performed, but the differences are not significant. In case n=2, please show the data as separate datapoints without error bars and statistics. See also: <http://www.embopress.org/page/journal/14693178/authorguide#statisticalanalysis>

If n<5, please show single datapoints for diagrams. Could statistics be provided for the diagrams shown in panels 1a-c and EV4b/c? It seems, some diagrams are presently missing the 'n.s.' (e.g. 6g-h). If n=2, please remove the error bars (EV2b/c?). Moreover:

- Please note that the legend for figure EV 2g is mislabeled as figure EV 2g, h in the manuscript. This needs to be rectified.
- Please note that the legends for figure EV 4a, b, c are mislabeled as figure EV 4c, a, b in the manuscript. This needs to be rectified.
- Please define the annotated p values ** as well as provide the exact p-values for the same in the legend of figure EV 1c; EV 2a-c, e; as appropriate.
- Please note that the exact p values are not provided in the legends of figures 1e-g; 2d, f; 3b-d, i; 4b, d, g, i; 6a-b, f-i; EV 4c, e.
- Please indicate the statistical test used for data analysis in the legends of figures 3i; 4d; 5a-f; EV 1c; EV 2a-c, e; EV 4c, e.
- Please note that axis gaps are not labeled appropriately in figure 1c.
- Please add to each legend (main and EV figures, where applicable) a 'Data Information' section explaining the statistics used or providing information regarding replicates and scales. See:

- The correct nomenclature for EV figures (figure labels, legends and callouts) is Figure EVx (instead of Extended View Figure x). Please fix this.
- Please make sure each figure panel is called out separately and sequentially. It seems presently callouts for EV figures 2, 4 and 5 are missing. Please check.
- Please make sure that all the funding information is also entered into the online submission system and that it is complete and similar to the one in the acknowledgement section of the manuscript text file. Presently the grants Wellcome Centre for Integrative Parasitology (Ref: 104111), the Lung Foundation Netherlands (project AWWA), NIH: grants R03 AI153915 and F30 AI157069, grants from the Beatson Institute and the Oncode institute base grants are missing in the submission system. Please check.
- Please provide the Appendix file as PDF. This needs to have page numbers in the ToC. The nomenclature of the Appendix Tables is Appendix Table Sx (not Appendix Table x). Please change the names and callouts accordingly.
- Please remove the author list and the example table from the reagents and tools table.

- Thank you for providing the requested source data. Please upload for each .prism file also a version in .csv format.

In addition, I would need from you:

Best,

Referee #1:

The authors have addressed all my questions, further revision is not needed.

Referee #2:

The authors have adequately addressed the concerns previously raised by this reviewer. The authors' amendments to the manuscript and/or clarification of the issues raised in my critique are appropriate.

All editorial and formatting issues were resolved by the authors.

Rick Maizels
University of Glasgow
School of Infection and Immunity
120 University Place
Glasgow G12 8TA
United Kingdom

Dear Dr. Maizels,

I am very pleased to accept your manuscript for publication in the next available issue of EMBO reports. Thank you for your contribution to our journal.

Please make sure that the dataset at the PRIDE repository is public latest upon online publication of the manuscript.

Yours sincerely,
